# Homo-BacPROTAC-induced degradation of ClpC1 as a strategy against drug-resistant mycobacteria

Lukas Junk [1,6] ✉, Volker M. Schmiedel [2,6], Somraj Guha[1], Katharina Fischel[2], Peter Greb [2], Kristin Vill [3], Violetta Krisilia [3], Lasse van Geelen[3], Klaus Rumpel [2], Parvinder Kaur [4], Ramya V. Krishnamurthy[4], Shridhar Narayanan [4], Radha Krishan Shandil[4], Mayas Singh[4], Christiane Kofink [2], Andreas Mantoulidis[2], Philipp Biber[2], Gerhard Gmaschitz[2], Uli Kazmaier[1], Anton Meinhart[5], Julia Leodolter [5], David Hoi[5], Sabryna Junker [5], Francesca Ester Morreale [5], Tim Clausen [5], Rainer Kalscheuer [3], Harald Weinstabl [2] ✉ & Guido Boehmelt [2] ✉

Antimicrobial resistance is a global health threat that requires the development of new treatment concepts. These should not only overcome existing resistance but be designed to slow down the emergence of new resistance mechanisms. Targeted protein degradation, whereby a drug redirects cellular proteolytic machinery towards degrading a specific target, is an emerging concept in drug discovery. We are extending this concept by developing proteolysis targeting chimeras active in bacteria (BacPROTACs) that bind to ClpC1, a component of the mycobacterial protein degradation machinery. The anti-*Mycobacterium tuberculosis* (*Mtb*) BacPROTACs are derived from cyclomarins which, when dimerized, generate compounds that recruit and degrade ClpC1. The resulting Homo-BacPROTACs reduce levels of endogenous ClpC1 in *Mycobacterium smegmatis* and display minimum inhibitory concentrations in the low micro- to nanomolar range in mycobacterial strains, including multiple drug-resistant *Mtb* isolates. The compounds also kill *Mtb* residing in macrophages. Thus, Homo-BacPROTACs that degrade ClpC1 represent a different strategy for targeting *Mtb* and overcoming drug resistance.

Based on the most recent analysis, bacterial antimicrobial resistance (AMR) is associated with 4.95 million deaths globally, with 1.97 million of those directly attributable to resistance[1]. Furthermore, AMR has been associated with prolonged hospital stays, increased healthcare costs, long-term disability, and loss of productivity[2,3]. Therefore, these data highlight that AMR is not only a global public health emergency, but a growing economic and societal burden. One of the reasons for the current AMR crisis is the dramatic slow-down in the development and approval of new antibiotics. The number of new approvals plunged from 63 new antibiotics approved between 1980 and 2000, to only 15 new ones approved between 2000 and 2018[4]. Additionally, the discovery of the last original class of antibiotics dates back to the late

[1]Organic Chemistry I, Saarland University, Campus Building C4.2, 66123 Saarbrücken, Germany. [2]Boehringer Ingelheim RCV GmbH & Co. KG, Dr. Boehringer-Gasse 5-11, 1121 Vienna, Austria. [3]Heinrich Heine University Düsseldorf, Faculty of Mathematics and Natural Sciences, Institute of Pharmaceutical Biology and Biotechnology, 40225 Düsseldorf, Germany. [4]Foundation for Neglected Disease Research, Plot No. 20A, KIADB Industrial Area, Veerapura Village, Doddaballapur, Bengaluru 561203 Karnataka, India. [5]Research Institute of Molecular Pathology, Vienna BioCenter, Vienna, Austria. [6]These authors contributed equally: Lukas Junk, Volker M. Schmiedel. ✉e-mail: lukasjunk0101@gmail.com; harald.weinstabl@boehringer-ingelheim.com; guido.boehmelt@boehringer-ingelheim.com

1980s[5]. Moreover, the concurrent rise of AMR has further diminished the effectiveness of the existing antibacterial drugs, highlighting the need to develop antibacterial agents that act via fundamentally new mechanisms.

In general, mechanisms that govern AMR are similar to those that diminish the effectiveness of anti-cancer treatments[6,7]. Therefore, some of the strategies used to combat drug resistance in cancer may also be implementable in the context of AMR. One such strategy is targeted protein degradation (TPD), which has recently emerged as a promising modality for targeted anticancer therapeutics[8]. Unlike traditional targeted therapeutics that usually exert their effects by inhibiting the target of interest, TPD agents redirect the activity of the cellular protein degradation machinery to degrade the target. A class of TPD agents that have been of growing interest are proteolysis targeting chimeras (PROTACs), small molecules that incorporate two ligands, one for the target of interest, and the other for an E3 ubiquitin ligase; the two ligands are connected via a linker of variable chemical composition and length to complete the PROTAC molecule[9]. All three components of the PROTAC molecule, the two ligands and the linker, play important functional roles to ensure that the target of interest is recruited to the E3 ubiquitin ligase machinery in a way that results in target poly-ubiquitination and subsequent degradation by the proteasome[10]. The approach allows various modifications of the general theme, like the reported proteasomal "self-degradation" of the E3 ligase adaptors von Hippel-Lindau (VHL) and cereblon by dimerized binders[11,12], or lysosome-mediated targeted degradation of extracellular proteins and membrane-associated proteins by LYTACs[13,14], as well as the selective removal of non-protein biomolecules or whole organelles by macroautophagy-targeting chimeras, MADTACs (including AUTACs and ATTECs)[15]. Given the degradation-based mechanism of PROTACs, they exhibit unique pharmacological properties, such as substoichiometric (catalytic) activity that improves efficacy and prolongs response[16], improved selectivity that minimizes off-target toxicity[17], activity against drug-resistant targets, and decreased likelihood of drug resistance[18]. The last two points suggest that a PROTAC-based antibacterial agent may overcome AMR.

However, bacteria don't have a ubiquitin-proteasome system. Instead, Gram-positive bacteria have evolved a recently discovered ClpC:ClpP:McsB system, whereby McsB acts as a "tagging kinase" that phosphorylates arginine residues in substrate proteins, which targets them to ClpC, an ATP-dependent unfoldase that recognizes and unfolds phospho-arginine (pArg) marked proteins and translocates them into the associated protease cage formed by ClpP units[19]. Thus, McsB could be considered as the prokaryotic counterpart to the eukaryotic ubiquitin E3 ligases, whereas pArg represents a small moiety equivalent of the poly-ubiquitin degradation tag. A recent proof-of-concept study validated that PROTAC-like molecules (BacPROTACs) redirect ClpC activity towards non-endogenous model protein substrates, resulting in their degradation via the ClpC:ClpP:McsB system[20]. In order to address whether BacPROTACs represent a viable strategy for antimicrobial drug development, we designed and tested a series of BacPROTACs directly aiming at the ClpCP machinery of *Mycobacterium tuberculosis* (*Mtb*), for two reasons:

First, *Mtb*, the causative agent of tuberculosis (TB), remains one of the deadliest human pathogens, leading to about 1.5 million TB deaths each year. Approximately 3% of new TB and 18% of previously treated TB cases are multidrug- or rifampicin-resistant (MDR/RR)[21]. Moreover, the COVID-19 pandemic has disproportionally affected TB services with the consequence that for the first time in a decade TB mortality has increased in 2021[22,23]. Therefore, developing new treatments for TB remains critically important[23]. Second, approaching the ClpCP system of mycobacteria is also guided by the fact that potent (nM active) binders, so-called cyclomarins, of ClpC1, the mycobacterium ClpC homolog, are described which should accelerate the first proof of concept experiments[24–28].

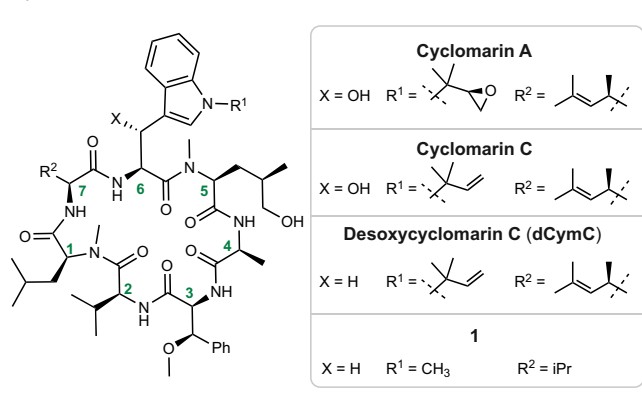

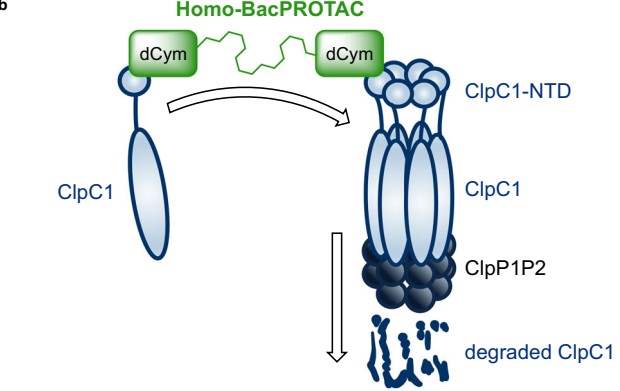

**Fig. 1 | Elements and mode of action of Homo-BacPROTACs. a** Naturally occurring cyclomarins and simplified synthetic dCym derivatives. The numbering system for amino acid positions used in this work is based on their introduction in the chemical synthesis and is indicated in the structure. **b** Concept for the Homo-BacPROTAC-induced degradation of ClpC1.

Here, we show the feasibility of developing BacPROTACs with anti-*Mtb* activity. Although *Mtb* does not have a homolog of McsB, it does have ClpC1, a homolog of ClpC, also functioning as a core element in the degradation machinery. We use cyclomarins as a chemical starting point for designing "Homo-BacPROTACs" that link two cyclomarin ligands together into a ClpC1-hijacking and targeting degrader molecule. We show that these molecules bind and degrade ClpC1 in vitro and remove a large portion of the intracellular pool of endogenous ClpC1 in a model mycobacterium, *M. smegmatis* (*Msm*). Importantly, these Homo-BacPROTACs exert potent bactericidal activity against both wildtype and multiple drug-resistant *Mtb* isolates, and also kill *Mtb* H37Rv residing in THP-1 macrophages. Taken together, our results suggest that Homo-BacPROTACs targeting *Mtb* ClpC1 for degradation provide a promising starting point for developing TPD-based strategies to thwart AMR.

## Results & Discussion
### Chemical synthesis of Homo-BacPROTACs

Using simplified desoxycyclomarin derivative **1** (Fig. 1a) as a backbone, we aimed to develop Homo-BacPROTAC molecules that simultaneously bind to two molecules of ClpC1. We hypothesized that the crosslinking of ClpC1 units would induce "self-degradation" of this essential unfoldase (Fig. 1b), leading to more efficient killing of mycobacteria as compared to the parental, non-degrading cyclomarins. Based on previous structure-activity relationship (SAR) studies[29], as well as the high-resolution X-ray crystal structure of ClpC1 N-terminal domain (NTD) bound to cyclomarin A (CymA)[24], we selected amino acids in positions 3, 6, and 7 as suitable exit vectors. The

**Fig. 2 | Chemical synthesis of Homo-BacPROTACs. a** Chemical synthesis of exit vector 6 Homo-BacPROTACs via CuAAC. **b** Synthesis of exit vector 7 Homo-BacPROTACs via CuAAC. **c** Synthesis of exit vector 6 and 3 Homo-BacPROTACs via olefin metathesis. BEP: 2-Bromo-1-ethylpyridinium tetrafluoroborate; DMBA: *N,N'*-Dimethylbarbituric acid; EDC: 1-Ethyl-3-(3-dimethylaminopropyl)carbodiimide hydrochloride; HOBt: 1-Hydroxybenzotriazole; NMM: *N*-methylmorpholine; TPPTS: 3,3',3''-Phosphanetriyltris(benzenesulfonic acid) trisodium salt; HATU: 1-[Bis(dimethylamino)methylene]-1H-1,2,3-triazolo[4,5-b]pyridinium 3-oxide hexafluorophosphate.

dimerization of the cyclic peptides was accomplished using either Cu(I)-catalyzed azide-alkyne cycloaddition (CuAAC) or olefin metathesis. The synthesis of exit vector 6 Homo-BacPROTACs started with literature-known pentapeptide **2** (Fig. 2). A sequence of two *N*-deprotections and peptide couplings with *N*1-propargylated tryptophan **A**[30] and allyloxycarbonyl-protected valine provided linear heptapeptide **4**, which was further converted to cyclomarin derivative **5** in a 4-step sequence. Using CuAAC reactions with different PEG- or alkyl

diazides, the corresponding dimers **6**, **7**, **8** (UdSBI-0545) and **9** were obtained (Fig. 2a). Two Homo-BacPROTACs bearing the exit-vector at amino acid 7 were synthesized from cyclic peptide **10** which was prepared analogously. The dimeric structures **11** and **12** (UdSBI-4377) were obtained by CuAAC reactions with 1,5-diazidopentane or 1,2-bis(2-azidoethoxy)ethane, respectively (Fig. 2b). By employing the enantiomeric starting materials, we additionally prepared enantiomeric Homo-BacPROTACs **8a** (UdSBI-0966) and **12a** (UdSBI-0117), to serve as distomeric negative controls in the degradation assays. The synthesis of two exit vector 3 Homo-BacPROTACs **SI-39** and **SI-40** was accomplished following the same synthetic route (see Supplementary Information). A propargyl ether attached to the *para*-position of β-methoxy-phenylalanine was used as the exit vector in this case. To obtain Homo-BacPROTACs with a more rigid linker which would also omit the polar triazole groups, we prepared cyclic heptapeptides **13** and **14** bearing either *O*- or *N*-allylated amino acids in positions 3 and 6. These compounds were then dimerized by olefin metathesis yielding Homo-BacPROTACs **15** and **17** (Fig. 2c). For **17**, however, we observed compound decomposition upon storage, which could also serve as an explanation for the low yield obtained in the synthesis of this dimer. Compound **16**, bearing a fully saturated linker was readily obtained by hydrogenation of the double bond in **15**.

In general, cyclic peptides are highly polar, which decreases their cellular permeability. We tested the cellular permeability of Cym derivatives as well as the Homo-BacPROTACs in Caco-2 cell monolayers and observed low permeability for all dimers tested (Supplementary Table 1). Therefore, we introduced peptide backbone modifications at solvent-exposed NH groups to increase the eukaryotic cell permeability. We identified the NH groups of valine and tryptophan as solvent-exposed, using nuclear magnetic resonance (NMR) methods[31,32] (Supplementary Tables 12, 13 and Supplementary Figs. 4–6). Since the NH of valine is involved in an H-bond network with ClpC1-NTD, we assumed that methylation of this amide would be detrimental to the binding affinity, and did not modify this position. Therefore, we focused on methylating the Trp-$N^\alpha$-H (pos. 6), which was solvent-exposed and not involved in protein binding. The synthesis of the Trp-$N^\alpha$-methylated derivatives was accomplished similarly to the other cyclomarin derivatives (Fig. 3). For the challenging couplings of *N*-methylated peptides, we in most cases used a slightly modified protocol introduced by Fuse et al.[33], employing catalytic amounts of *N*-methylimidazole and HCl. We observed lower yields in the cyclization reactions of heptapeptides such as **19** than in the case of non-methylated peptides, which we attribute to a less favorable conformation in Trp-$N^\alpha$-methylated peptides. When conducted at higher temperatures in 1,2-dichloroethane, the cyclizations however yielded the *N*-methylated compounds in synthetically useful yields. The homodimerizations of these derivatives were accomplished using the same conditions as described above.

## Structure-Activity Relationship (SAR) study of ClpC1-NTD targeting Homo-BacPROTACs

To characterize Homo-BacPROTACs and compare them to the monomeric derivatives, we used surface plasmon resonance (SPR) to measure the dissociation constants ($K_D$) towards ClpC1-NTD. We observed that Homo-BacPROTACs displayed an improved binding affinity for the NTD of ClpC1, roughly by a factor of 10 when compared to monomeric compounds (Table 1, Supplementary Table 1). To investigate whether Homo-BacPROTACs bind monovalently to the ClpC1-NTD surface, we performed a stacking experiment (Supplementary Fig. 210). When injecting ClpC1-NTD over a surface of biotinylated ClpC1-NTD saturated with dimeric Homo-BacPROTAC, we observed very limited binding (10% of the expected response), suggesting that only a fraction of the presumably free Cym moieties of the Homo-BacPROTAC were available for binding to the injected protein. The simplest explanation for this result is that the Homo-BacPROTAC

is bound bivalently to different ClpC1-NTD molecules on the chip surface.

Along those lines, weaker binding was observed for some Homo-BacPROTACs with rigid linkers, whereas the influence of linker length upon binding was neglectable. The introduction of the methyl group at amino acid 6 led to weaker binding, which remained in the single-digit nanomolar range (Table 1). Exceptions to the latter observation were the two matched pairs **8/27** (entry 6) and **15/28** (entry 9), in which the matched pairs bind to ClpC1-NTD with nearly equal strength. The intended improved permeability into Caco-2 cells for Trp-$N^\alpha$-methylated derivatives could clearly be observed for monomeric compounds (Table 1, full details in Supplementary Table 1). The alkyne-containing dimerization precursors **20** and **26** showed efflux ratios of 11.8 and 2.2, thereby being more permeable by roughly one order of magnitude compared to their matched pairs lacking the alkylation (entries 2 and 5). The effect was slightly diminished when comparing dCymC to its alkylated counterpart **23** (entry 1) and model compounds **24/25**, bearing n-pentyl-triazolyl motifs at the linker attachment points (entry 3). In these cases, N-methylation led to a 1.5 times increase in permeability, respectively. Unfortunately, permeability of Homo-BacPROTACs proved difficult to measure in our hands using a variety of assay conditions. Aqueous solubility remained low throughout the compounds synthesized in this study and could only be improved upon the introduction of more polar side chains. These side chains interfered with compound binding to ClpC1-NTD and were therefore discarded in further designs.

We also measured metabolic stability and saw improvements over the monomeric compounds (full details in Supplementary Table 2), which might be due to oxidative liability of unsaturated side chains in these compounds that are absent from our Homo-BacPROTACs. This improvement in in vitro metabolic stability was reflected in an in vivo PK study of three Homo-BacPROTACs and one monomer (BALB/cAnNCrl mice, 1 mg kg$^{-1}$ bolus IV administration): **5**, **6**, **8** and **12**. The Homo-BacPROTACs **6** and **8**, which use position 6 (Trp) as exit vector, showed higher $c_{max}$ and AUC, but lower clearance than monomer **5** (Supplementary Table 3). The monomer **5** was metabolized and excreted quickly and was unable to build up meaningful levels in blood. A corresponding oral PK study of these compounds did not result in an interpretable plasma exposure, due to very slow adsorption in the gut caused by the low solubility of the compounds (Supplementary Tables 14–17). Taken together, Homo-BacPROTACs outperformed the parental cyclic peptides in terms of binding affinity and metabolic stability, although additional optimization of these compounds is needed to obtain cellular permeability and oral bioavailability.

## Homo-BacPROTACs are potent degraders of ClpC1-NTD

Since the affinity towards ClpC1-NTD of various Homo-BacPROTACs described above is in the sub-nM range, we assessed whether such molecules could induce degradation of their cognate target ClpC1 in a cell-free degradation assay. To this end, a minimal recombinant degradation machinery consisting of full-length *Msm* ClpC1, processed *Msm* ClpP1 and ClpP2, combined with an ATP-regenerating system[20], was adapted to allow quantifiable capillary Western (WES) readout (Supplementary Table 8 and Supplementary Fig. 3). We first used His$_6$-tagged ClpC1-NTD as substrate and analyzed various Homo-BacPROTACs with respective controls. As shown in Fig. 4, Homo-BacPROTAC **8** (UdSBI-0545) induces efficient removal of ClpC1-NTD substrate with a half-maximal degradation concentration (DC$_{50}$) of 8.0 μM and a maximum degradation efficacy (D$_{Max}$) of 83%. Similarly, Homo-BacPROTAC **12** (UdSBI-4377) induces degradation with a DC$_{50}$ of 8.4 μM and a D$_{Max}$ of 81% (Fig. 4a, b and Supplementary Table 8). In contrast, vehicle, dCymC, the matching monomers **5** (UdSBI-6231) and **10** (UdSBI-5602), or Homo-BacPROTAC enantiomers **8a** (UdSBI-0966) and **12a** (UdSBI-0117), did not cause any degradation of the His$_6$-tagged ClpC1-NTD substrate (Fig. 4a–d and Supplementary Table 8).

**Table 1 | Structure-activity relationships of monomeric dCym derivatives and Homo-BacPROTACs**

| Entry | Compound | | R1 | R2 | R3 | ClpC1-NTD KD[a] [nM] Mean ± SD | MIC Mtb H37Rv[b] [μM] Mean ± SD | Influence of N-methylation on permeability[c] |
|---|---|---|---|---|---|---|---|---|
| 1 | dCymC | Monomer | H | | | 1.1 ± 0.4 | 0.4 ± 0.1 | |
| | **23** | | Me | (allyl) | (prenyl) | 12.1 ± 0.3 | n.d. | 1.6 |
| 2 | **10** | Monomer | H | Me | (propargyl) | 4.0 ± 0.7 | 3.1 ± 0.0 | >11.8 |
| | **20** | | Me | | | 13.9 ± 0.3 | 50.0 ± 0.0 | |
| 3 | **24** | Monomer | H | Me | A | 4.6 ± 0.2 | n.d. | >1.5 |
| | **25** | | Me | | | 26.1 ± 0.1 | 6.3 | |
| 4 | **12** (UdSBI-4377) | Dimer | H | Me | B | 0.28 ± 0.08 | 0.1 ± 0.0 | n.d. |
| | **22** | | Me | | | 0.9 ± 0.3 | 3.1 ± 0.0 | |
| 5 | **5** | Monomer | H | (propargyl) | iPr | 3.5 ± 0.1 | 1.6 | 9.3 |
| | **26** | | Me | | | 15 ± 3 | 6.3 | |
| 6 | **8** (UdSBI-0545) | Dimer | H | E | iPr | 0.4 ± 0.1 | 0.4 ± 0.4 | n.d. |
| | **27** | | Me | | | 0.4 ± 0.2 | <0.1 | |
| 7 | **6** | Dimer | H | C | iPr | 0.9 ± 0.4 | 0.4 ± 0.3 | – |
| 8 | **11** | Dimer | H | Me | D | 0.4 ± 0.2 | 0.1 ± 0.0 | – |
| 9 | **15** | Dimer | H | (butenyl) | iPr | 1.2 ± 0.8 | 0.2 ± 0.1 | n.d. |
| | **28** | | Me | | | 1.8 ± 0.1 | n.d. | |

$K_D$ and MIC values are reported as mean ± SD. The number of measurements performed for each compound are indicated by footnotes in Supplementary Tables 1 and Table 2.
[a]Dissociation constant $K_D$ towards ClpC1-NTD measured by surface plasmon resonance.
[b]Minimum inhibitory concentrations against *M. tuberculosis* H37Rv strain.
[c]Determined by comparison of efflux ratios across Caco-2 monolayers for matched pairs: $R_{eff}$ (N-H)/$R_{eff}$ (N-Me). n.d.: not determined. Bold numbers refer to compounds described in the text.

We also tested whether the Homo-BacPROTACs would be comparably active in a reconstituted degradation assay in which the three *Msm* proteins were replaced by their corresponding *Mtb* homologues. To this end, various Homo-BacPROTACs, their distomers, and matching monomeric compounds were tested in the *Mtb* cell-free degradation assay and we observed Homo-BacPROTAC-mediated degradation of the ClpC1-NTD (Supplementary Table 8), with comparable $DC_{50}$ values and slightly reduced $D_{Max}$ values as seen for *Msm* ClpC1P1P2. As expected, the control compounds did not cause degradation.

We also analyzed whether full-length ClpC1 would be degraded in this assay, by analyzing the stability of the full-length form monitored by a ClpC1-N-terminal antibody in the WES assays, or directly by SYPRO™ Ruby stained SDS-PAGE gels in the presence or absence of ClpC1-NTD as additional substrate. The Homo-BacPROTACs tested induced no or only minor degradation of the full-length ClpC1 under these in vitro conditions (Fig. 4e). Whether this is due to hexamer formation stabilizing ClpC1 protomers or missing cofactor(s) of the protease complex, is currently unclear. Likewise, the exact sequential and structural features of native substrates targeted by ClpC1 are not known, except that partially disordered peptide segments can promote degradation[20].

### Homo-BacPROTACs induce degradation of endogenous ClpC1 in *Msm*

Irrespective of the behavior of Homo-BacPROTACs in cell-free assay systems, we wanted to assess their potential to degrade endogenous ClpC1 in *Msm*. Using a capillary Western-based cellular degradation assay, we could show that Homo-BacPROTAC **8** (UdSBI-0545) induced degradation with an average $DC_{50}$ of $0.57 ± 0.40$ μM and an average $D_{max}$ of $48 ± 13\%$ after 24 h of incubation in a concentration-dependent manner (Fig. 5a), while the enantiomeric control **8a** (UdSBI-0966) did not lead to intracellular degradation of ClpC1 (Fig. 5b). Homo-BacPROTAC **12** (UdSBI-4377) also induced full-length ClpC1 degradation (average $DC_{50}$ $0.17 ± 0.10$ μM; average $D_{max}$ $43 ± 8\%$), while its enantiomer **12a** (UdSBI-0117) was inactive (Fig. 5b). In addition, the matching monomers **5**, **10** and dCymC did not induce degradation of ClpC1 (Fig. 5c). Consistent with our findings, recent proteomics analyses highlighted that although monomeric cyclomarin derivatives and Homo-BacPROTACs both bind to ClpC1 causing strong perturbations in the mycobacterial proteome, only Homo-BacPROTACs induce efficient degradation of ClpC1 in *Mtb* at concentrations and time points consistent with their activity in MIC assays[34]. Together, these two

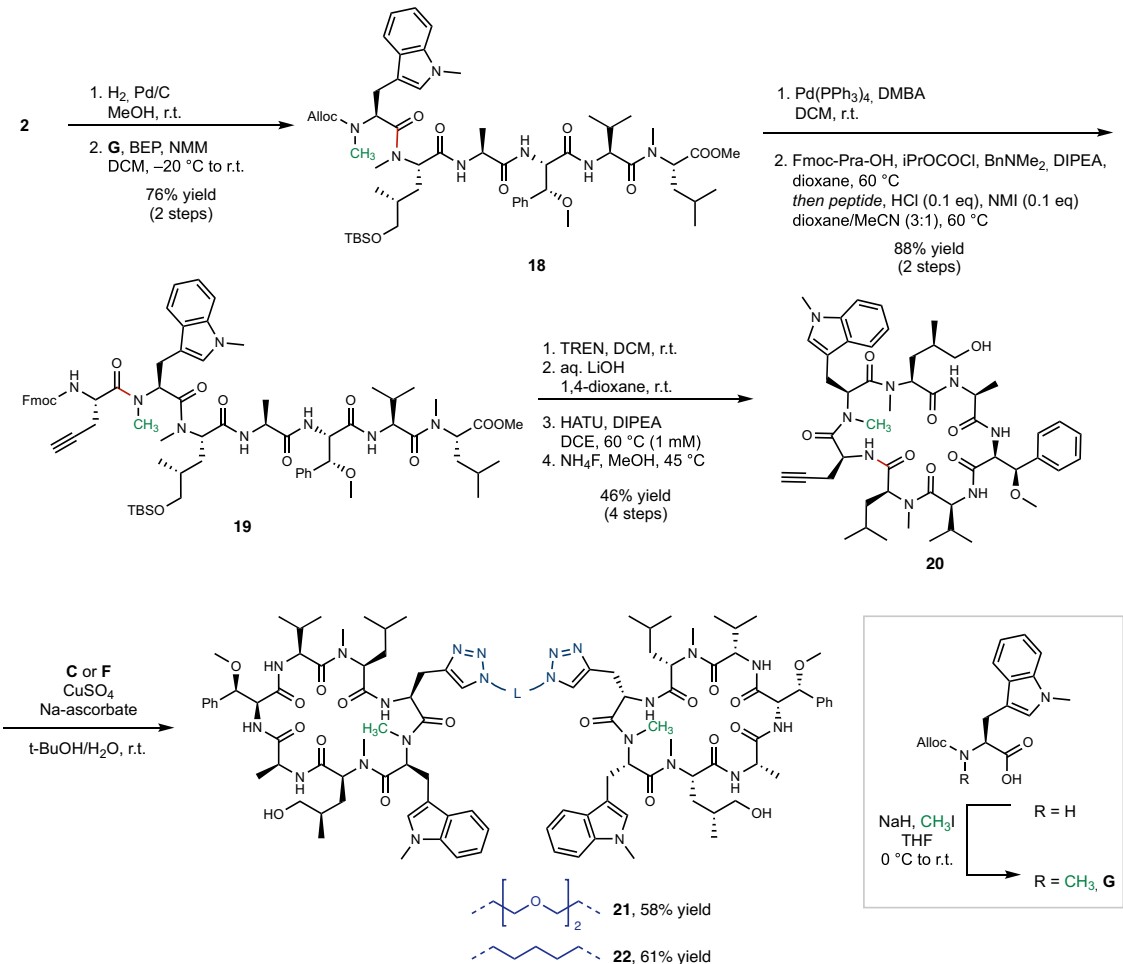

**Fig. 3 | Further modifications of exit vector 7 Homo-BacPROTACs.** Chemical synthesis of exit vector 7 Homo-BacPROTACs bearing an additional methyl group (shown in green) attached to tryptophan-N$^\alpha$. Pra Propargylglycine. NMI *N*-methylimidazole. TREN Tris(2-aminoethyl)amine.

studies thus present the first example of Homo-BacPROTAC-mediated targeted degradation of an endogenous protein in mycobacteria. Several reasons could compromise the cellular activity of Homo-Bac-PROTACs, preventing complete degradation of their target in *Msm*. Such parameters could include the potentially smaller permeability of these dimeric molecules through bacterial membranes, putative resistance mechanisms, and altered half-life or re-synthesis rate of ClpC1. In fact, further analysis of the moderate degradative efficacy of the Homo-BacPROTACs in *Msm* led to the discovery of an antibiotic scavenger system protecting ClpC1 and the ClpC1P1P2 protease[34].

### Biological activity of Homo-BacPROTACs

We further investigated, whether Homo-BacPROTACs would be active in inhibiting the proliferation of pathogenic mycobacteria, and how their cellular activity compares to their matched monomeric compounds or the parental dCymC. We, therefore, assessed the minimum inhibitory concentrations (MICs) against *Mtb* H37Rv in liquid culture resazurin assays. While dCymC inhibits the growth of *Mtb* H37Rv at $0.4 \pm 0.1\,\mu M$, the alkyne-bearing monomeric derivative **10** does so at $3.1 \pm 0.0\,\mu M$. Interestingly, the corresponding Homo-BacPROTAC **12** (UdSBI-4377) inhibits the growth of this *Mtb* strain at $0.1 \pm 0.0\,\mu M$. In this series, the SPR $K_D$ to ClpC1-NTD improves by a factor of 10 from monomeric **10** to dimeric **12**, whereas the MICs for these compounds against *Mtb* improve by a factor of 30. This can be seen as an indication of the catalytic mode of action of Homo-BacPROTACs, since similar improvements in MIC potency on *Mtb* H37Rv by Homo-BacPROTACs

as compared to their matching monomeric counterparts were also observed for other series (Supplementary Table 2).

We also tested various Homo-BacPROTACs and control compounds for their MIC on *Msm #607*. Interestingly, the MIC values were higher than those observed for *Mtb* H37Rv and, in addition, very comparable between Homo-BacPROTACs and matching monomers (Supplementary Table 2). This indicates that these two strains of mycobacteria respond differently to Homo-BacPROTACs. Consistently, quantitative proteomics demonstrated the higher efficacy of Homo-BacPROTACs in inducing ClpC1 degradation in the *Mtb* H37Rv strain as compared to *Msm*[34].

To further confirm the qualitative difference in responsiveness between *Msm* and *Mtb*, we assessed whether Homo-BacPROTACs are bactericidal or bacteriostatic. To this end we performed MBC (minimum bactericidal concentration) assays, where, following a MIC assay, compound-treated cultures were replated on 7H9 complete agar in the absence of compound. While growth of colonies could be observed across all tested concentrations (up to 50 µM) of **8** (UdSBI-0545) or **12** (UdSBI-4377) in the case of *Msm #607*, *Mtb* cultures treated with these Homo-BacPROTACs did not form any colonies at concentrations at or above the MIC value. These results are consistent with the notion that Homo-BacPROTACs are bacteriostatic on *Msm #607*, but bactericidal on *Mtb* H37Rv (Supplementary Table 11).

Next, we analyzed whether Homo-BacPROTACs would be selective for mycobacteria, as one would anticipate based on the limited homology within ClpC family members across prokaryotes.

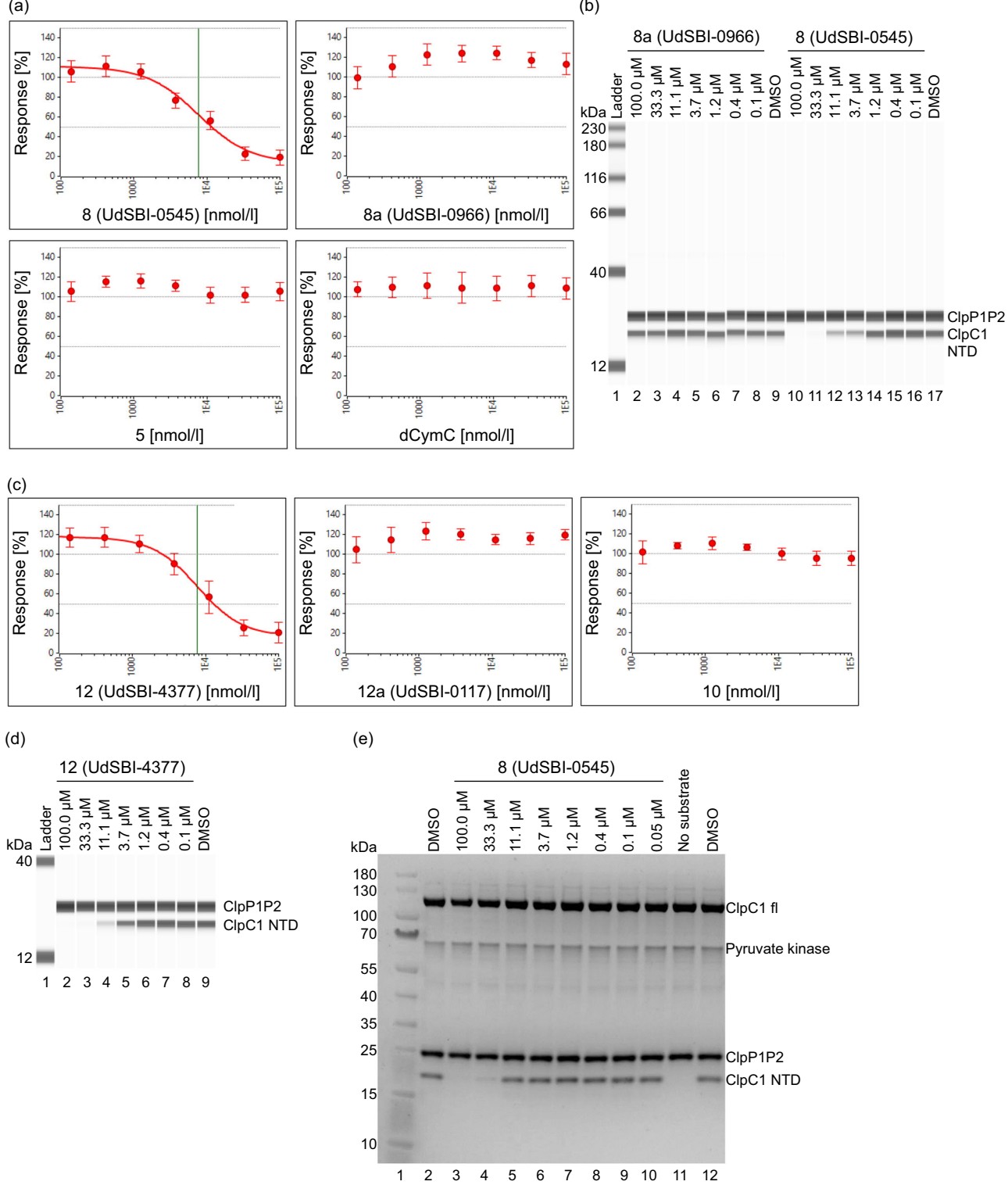

**Fig. 4 | Activity of Homo-BacPROTACs in cell-free degradation assays.**
**a** Degradation curves of ClpC1-NTD in the cell-free degradation assay (quantified by capillary Western (WES)) induced by Homo-BacPROTAC **8** (UdSBI-0545) compared to its enantiomer **8a** (UdSBI-0966), matching monomer **5** and dCymC (3 independent experiments done in triplicates). **b** WES visualization of ClpC1-NTD degradation after titration of Homo-BacPROTACs, developed with anti-His antibody recognizing His6-tagged ClpC1-NTD and processed, His4-tagged ClpP1P2. Concentration-dependent degradation of ClpC1-NTD can be observed for **8** (UdSBI-0545) (lanes 10–13) but not for **8a** (UdSBI-0966) (lanes 2–8). **c** Analogous to **a**, except that exit vector 7-based Homo-BacPROTAC **12** (UdSBI-4377), enantiomer

**12a** (UdSBI-0117) and monomer **10** were used. **d** WES-derived gel picture visualizing ClpC1-NTD degradation (lane 2–5) from representative experiment summarized in **c**. **e** SYPRO™ Ruby-stained SDS-PAGE gel from exemplary cell-free degradation assay depicting efficient degradation (lane 3,4) of ClpC1-NTD by Homo-BacPROTAC **8** (UdSBI-0545), while full length ClpC1 is not significantly degraded. Green vertical lines indicate the DC$_{50}$ (for **8, 12**). Error bars represent mean ± SD of $n = 3$ independent experiments in triplicates. The actual mean DC$_{50}$ values for all cell-free degradation experiments conducted for this study are summarized in Supplementary Table 8. Source data are provided as a Source Data file.

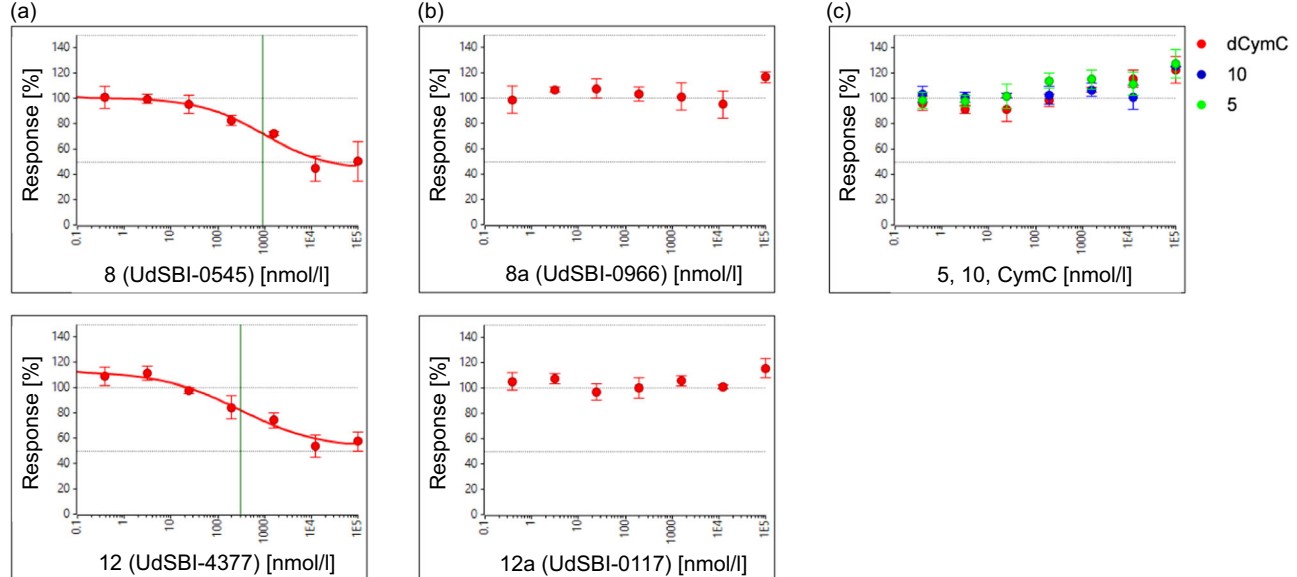

**Fig. 5 | Homo-BacPROTACs degrade intracellular ClpC1. a** Intracellular degradation of ClpC1 in *Msm #607* cells following a 24 h incubation with Homo-BacPROTACs **8** (UdSBI-0545), or **12** (UdSBI-4377). Green vertical lines indicate the $DC_{50}$ values obtained in that particular experiment. **b** ClpC1 levels in *Msm #607* following a 24 h incubation with distomers **8a** (UdSBI-0966), or **12a** (UdSBI-0117). **c** ClpC1 levels in *Msm #607* following a 24 h incubation with monomers **5** (green), **10** (blue), or dCymC (red). The curves show representative experiments performed in well triplicates. Experiments for each compound were performed independently $n = 3$ (compounds **8, 8a, 12a, dCymC, 5, 10**) or $n = 4$ (compound **12**) times with similar results. Error bars indicate mean $\pm$ SD of $n = 3$ well replicates for that given experiment. The actual mean $DC_{50}$ values reported in the text for compound **8** or **12** are calculated from the respective individual $DC_{50}$ values obtained for the independent cellular degradation experiments conducted in this study and are summarized in the Source Data file for Fig. 5a. Source data are provided as a Source Data file.

To this end, various Gram positive and Gram negative bacteria were tested in MIC assays using Homo-BacPROTACs **8** (UdSBI-0545), **12** (UdSBI-4377), and their matching monomers **5** and **10**, respectively. As expected, no inhibition of growth was observed (Supplementary Table 9). To obtain further resolution of potency within the family of mycobacteria, various disease-associated representatives like *M. avium*, *M. abscessus*, *M. fortuitum*, and *M. intracellulare* were assessed. Interestingly, at the tested concentrations, no growth inhibition could be observed for the two Homo-BacPROTACs for these NTM (nontuberculous mycobacteria) strains, while the matching monomeric compounds were active in the single-digit μM range (Supplementary Table 10).

Due to the potent bactericidal activity of these Homo-BacPROTACs towards the well studied *Mtb* H37Rv strain, and the assumption that Homo-BacPROTACs would also be active against drug resistant strains, we investigated whether various *Mtb* isolates, including isolates being resistant to known TB drugs, respond to incubation with Homo-BacPROTACs **8** (UdSBI-0545), or **12** (UdSBI-4377). This includes *Mtb* isolates like the RpoB S450L mutant conferring rifampicin resistance, the *katG* deletion strain, conferring isoniazid resistance, a clincial isolate (FNDR-M1) resistant to moxifloxacin, or the Beijing strain HN878. As can be seen in Table 2, Table 3 and Supplementary Fig. 1, the Homo-BacPROTACs were highly potent on such isolates and outperformed the matching monomers.

Next, we asked to what extent non-growing *Mtb* H37Rv cells would be susceptible to Homo-BacPROTACs, matching monomers or standard-of-care drugs like bedaquiline, D-cycloserine, rifampicin, ethambutol and moxifloxacin. Starvation for three weeks was chosen as a mechanism to induce the non-growing state of the *Mtb* H37Rv cells prior to addition of compounds. Homo-BacPROTACs **8** (UdSBI-0545), or **12** (UdSBI-4377) lost their inhibitory activity similar to several of the TB drugs, while bedaquiline, rifampicin and the matching monomers retained a partial inhibitory activity (Supplementary Fig. 2).

Along those lines, and to assess other dormancy mimicking states, Homo-BacPROTACs were recently tested on Mtb H37Rv cells in combination with bedaquiline[34]. By inhibiting the ATP synthase subunit C (AtpE), bedaquiline induces lower intracellular ATP levels, resembling a dormant state in *Mtb*. The presence of bedaquiline had no effect on the activity of **12** (UdSBI-4377) against *Mtb* H37Rv[34]. Homo-BacPROTAC activity may thus depend under some circumstances on the environmental conditions presented to the *Mtb* cells.

Lastly, pathogenic *Mtb* strains reside in eukaryotic cells including macrophages, which is part of their immune system-evading strategy leading to an indefinite persistence in infected individuals. This may eventually lead to reactivation and thereby the emergence of new tuberculosis cases[36,37]. We, therefore, asked whether the Homo-BacPROTACs can impair the survival of *Mtb* H37Rv in macrophages differentiated from the monocytic leukemia cell line THP-1. We compared the effects of Homo-BacPROTACs to those of their corresponding monomers and the reference antibiotics rifampicin and moxifloxacin, which are known to inhibit the intracellular propagation of *Mtb*. Maximal concentrations of compounds employed in these studies were pretested not to affect uninfected, differentiated THP-1 cells. Exit vector 6 Homo-BacPROTACs clearly showed a concentration-dependent reduction of colony-forming units (cfu) over time with an $E_{max}$ of 1.22–1.27 at 50 μM (Fig. 6a), while exit vector 7 Homo-BacPROTACs were less efficient with an $E_{max}$ ranging from 0.28–0.51 at the same concentration (Fig. 6b). Homo-BacPROTACs performed more efficiently over time than their matched monomeric counterparts (Fig. 6a), or reduced intracellular *Mtb* at lower concentrations (Fig. 6b). These results demonstrate that Homo-BacPROTACs can enter eukaryotic host cells and kill intracellularly residing pathogenic mycobacteria.

In conclusion, we herein describe the synthesis and characterization of Homo-BacPROTACs. These compounds are composed of two cyclic heptapeptides derived from the natural product class of cyclomarins and induce degradation of mycobacterial ClpC1, a

**Table 2 | MICs of Homo-BacPROTACs and monomers against clinical isolates resistant to single *Mtb* drugs**

| Compound | *Mtb* H37Rv | *Mtb* (moxR) FNDR-M1 | *Mtb* (RpoB S450L) ATCC 35838 MIC (µM) | *Mtb* (*katG* del) ATCC 35822 |
|---|---|---|---|---|
| **12** (UdSBI-4377) | 0.1 ± 0.0 | 0.2 ± 0.0 | 0.2 ± 0.0 | 0.1 ± 0.0 |
| **8** (UdSBI-0545) | 0.2 ± 0.1 | 0.2 ± 0.0 | 0.2 ± 0.0 | 0.2 ± 0.0 |
| **10** | 3.1 ± 0.0 | 1.6 ± 0.0 | 1.6 ± 0.0 | 1.6 ± 0.0 |
| **5** | 2.6 ± 0.9 | 0.7 ± 0.2 | 0.4 ± 0.0 | 0.8 ± 0.0 |
| Rifampicin | 0.01 ± 0.0 | 0.01 ± 0.0 | **>38.9** | 0.01 ± 0.0 |
| Moxifloxacin | 0.2 ± 0.0 | **19.9 ± 0.0** | 0.3 ± 0.0 | 0.3 ± 0.0 |
| Isoniazid | 0.2 ± 0.0 | 0.1 ± 0.0 | 0.5 ± 0.0 | **>232.2** |

Bold numbers refer to compounds described in the text. MIC values reflecting resistance are indicated in bold.
Values represent means and standard deviations from $n = 3$ well replicates.

**Table 3 | Homo-BacPROTACs are potent on multiple drug-resistant *Mtb* strains**

| Compound | MIC *Mtb* # 11291 | | MIC *Mtb* # 8673 | | MIC *Mtb* ATCC 35825 | |
|---|---|---|---|---|---|---|
| | µM | µg/ml | µM | µg/ml | µM | µg/ml |
| **10** | 0.4 | 0.4 | 0.4 | 0.4 | 0.4 | 0.4 |
| **12** (UdSBI-4377) | <0.1 | <0.2 | <0.1 | <0.2 | <0.1 | <0.2 |
| **5** | 0.2 | 0.2 | 0.2 | 0.2 | 0.2 | 0.2 |
| **8** (UdSBI-0545) | <0.1 | <0.2 | <0.1 | <0.2 | <0.1 | <0.2 |
| Moxifloxacin | 0.2 | 0.1 | 0.1 | 0.1 | 0.1 | 0.1 |
| Streptomycin | 0.4 | 0.3 | 6.9 | 4.0 | **27.5** | **16.0** |
| Isoniazid | 3.6 | 0.5 | **466.7** | **64.0** | **29.2** | **4.0** |
| Rifampicin | 0.005 | 0.004 | **38.9** | **32.0** | 0.005 | 0.004 |
| Amikacin | 3.4 | 2.0 | **>218.6** | **>128.0** | 0.1 | 0.03 |
| Thiacetazone | 4.2 | 1.0 | **>541.7** | **>128.0** | 4.2 | 1.0 |
| p-Aminosalicylic acid | 6.5 | 1.0 | 6.5 | 1.0 | **104.5** | **16.0** |

Bold numbers refer to compounds described in the text. MIC values reflecting resistance are indicated in bold.
Values represent means from $n = 2$ well replicates.

component of the bacterial proteolytic machinery. The dimeric compounds are able to redirect the Clp protease against its own ClpC1 subunit, ultimately leading to the "self-destruction" of the essential unfoldase in mycobacteria. We show that Homo-BacPROTACs cause the degradation of ClpC1-NTD in vitro and the degradation of endogenous full-length ClpC1 in Msm cells. Moreover, compared to monomeric counterparts, improved activity against Mtb wild-type cultures, Mtb residing within macrophages and drug-resistant strains is observed.

However, the reported Homo-BacPROTACs have several limitations, primarily with respect to their physicochemical properties, such as solubility and total polar surface area. This limits their bioavailability and pharmacodynamic profiling in vivo and will require further optimization. Our preliminary SAR studies describe potential strategies going forward and opportunities for synthesis of the next generation of Homo-BacPROTACs. In addition, some nontuberculous mycobacteria do not respond to the Homo-BacPROTACs which currently prevents their use in other disease types. A possible explanation could be especially high or upregulated levels of the counteracting ClpC2 and/or ClpC3 proteins, which were shown to dampen the efficacy of Homo-BacPROTACs in *Msm* as compared to *Mtb* cells[35]. Multiple compound-specific parameters such as metabolic stability, bacterial

uptake and efflux behavior could also contribute to different responses in NTM strains.

In summary, our study provides the first example of BacPROTACs that induce degradation of an endogenous bacterial protein, which could serve as a blueprint for the development of anti-mycobacterial agents that may eventually overcome antimicrobial resistance.

## Methods

### Protein Expression and Purification

Full-length *M. smegmatis* ClpC1 (untagged), ClpP1 (C-terminal His$_4$-tag) and ClpP2 (C-terminal His$_4$-tag) proteins were cloned, expressed and purified as described[20].

Full-length *M. tuberculosis* ClpC1 (untagged), ClpP1 (C-terminal His$_4$-tag) and ClpP2 (C-terminal His$_4$-tag) proteins were cloned, expressed and purified as described[37].

N-terminal propeptides of ClpP1 and ClpP2 were processed with the activator peptide Z-Leu-Leu-H (Benzyloxycarbonyl-L-Leucyl-L-Leucinal) as described[37].

Cloning, expression and purification of *M. tuberculosis* ClpC1 NTD (AA 1-148, UniProt P9WPC9) with a C-terminal non-cleavable hexa-histidine tag was based on the protocol described for its homolog ClpC from *B. subtilis*[19], with the following modifications:

Cells were lysed by sonication in 50 mM Tris, pH 8.0, 150 mM NaCl. The cleared lysate was applied to XK16 Ni-NTA beads (Cytiva) and eluted with lysis buffer containing 250 mM imidazole. The protein was further purified by size exclusion chromatography in 50 mM Tris, pH 8.0, 150 mM NaCl on a HiLoad 16/600 Superdex 200 pg column (Cytiva) and ClpC1 NTD containing fractions were pooled. Sequence of the expressed protein:

MFERFTDRARRVVVLAQEEARMLNHNYIGTEHILLGLIHEGEGVAAK SLESLGISLEGVRSQVEEIIGQGQQAPSGHIPFTPRAKKVLELSLREALQLGH NYIGTEHILLGLIREGEGVAAQVLVKLGAELTRVRQQVIQLLSGYQGKLEHH HHHH

All proteins were aliquoted and stored at −80 °C until further use.

### SPR binding studies

SPR experiments were performed on T200 and 8k instruments (Cytiva).

ClpC1 NTD protein (*M. tuberculosis*) was chemically biotinylated using an EZ-Link NHS-PEG4-biotin kit (Thermo Scientific). The labelling procedure was performed according to the manufacturer's instructions with the following modifications: A 1:5 molar ratio of biotin reagent to protein was used and the reaction was incubated for 3 h at room temperature under agitation in 10 mM HEPES, pH 7.4, 150 mM NaCl, 0.05 % Tween20 (HBS-P+). Non-reacted NHS-PEG4-Biotin was subsequently removed using a Zeba Spin desalting column (Thermo Scientific).

Streptavidin (Prospec) was immobilized in 10 mM Na acetate, pH 5.0 at a density of 2000 – 3000 RUs onto flow cells 1 and 2 of a CM5 chip (Cytiva) at 25 °C using an amine-coupling kit (Cytiva) under standard conditions. Biotinylated ClpC1 was then captured to flow cell 2 at a density of 100–200 RUs in HBS-P + .

Compound binding was subsequently analyzed at 25 °C in 25 mM Tris(hydroxymethyl)aminomethane, pH 7.5, 150 mM NaCl, 0.01% Tween20, 1% dimethyl sulfoxide.

Sensorgrams were recorded at different compound concentrations in single-cycle mode and double-referenced prior to data analysis using Biacore Insight software. Data were fitted using the 1:1 interaction model with a term for mass-transport included. A few compounds showed deviation from a mono-exponential decay in the form of a slow-off / irreversible component in the later parts of the dissociation phase. It turned out that for these compounds the U-value (uniqueness-value), a statistical fit quality parameter calculated by the Insight software, was unacceptably high. In these cases, the latter part of the dissociation curve was not included in the fit. This resulted in a

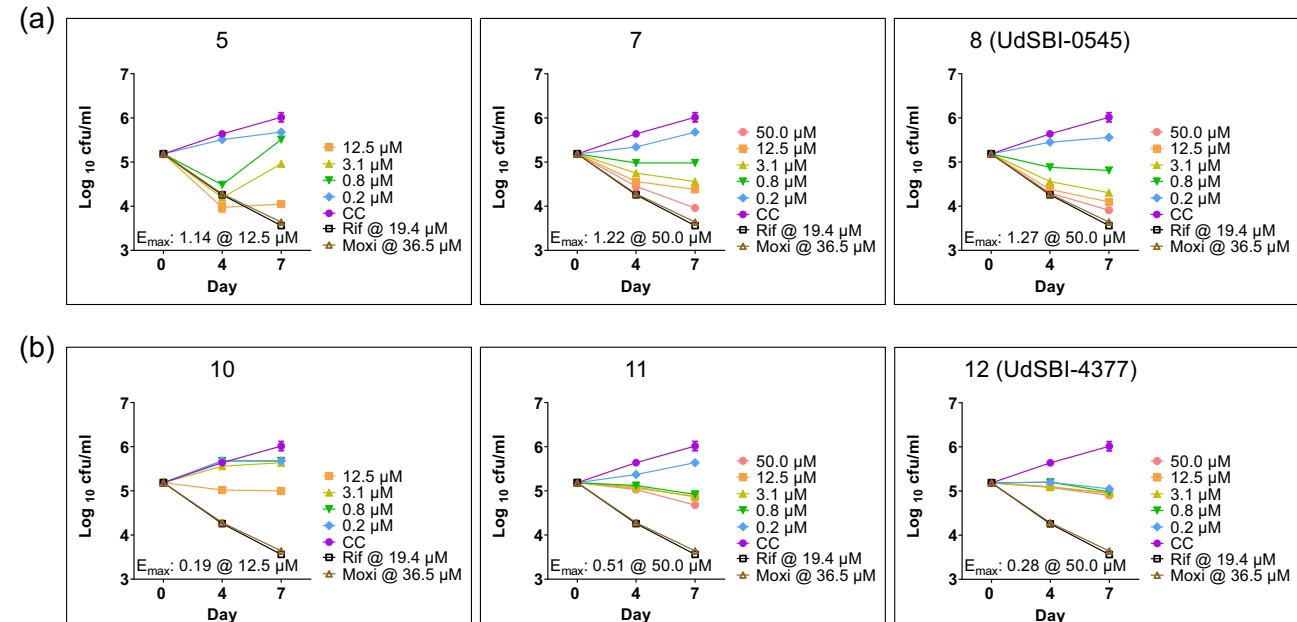

**Fig. 6 | Intracellular MIC assay in THP-1 macrophages. a, b** Exit vector 6-based Homo-BacPROTACs **7** and **8** (UdSBI-0545), as well as exit vector 7-based Homo-BacPROTACs **11** and **12** (UdSBI-4377) were assessed in a 4- and 7-day incubation on THP-1 macrophages following infection with *Mtb* H37Rv. The Homo-BacPROTACs were compared to their corresponding monomers (**5** for exit vector 6 and **10** for exit vector 7) and the reference antibiotics rifampicin and moxifloxacin, which are known to inhibit the intracellular propagation of *Mtb* H37Rv. Exit vector 6 Homo-BacPROTACs showed a more efficient concentration-dependent reduction of cfu/ ml over time with an $E_{max}$ of 1.22 − 1.27 at 50 μM, as compared to exit vector 7 based Homo-BacPROTACs or matched monomers. The various compound concentrations are indicated in different colours, matching across the panels. CC means *Mtb* culture control, where no drug treatment was given. For further details see text. Error bars indicate mean ± SD of $n = 2$ well replicates. The CC, rifampicin and moxifloxacin values were taken as common reference points in the various graphs. Source data are provided as a Source Data file.

significantly improved fit quality as judged by the U-value and visual comparison of sensorgram and fit.

At least two independent measurements were performed for each compound with mean values and standard deviations of these independent measurements reported. The same molar concentrations were used for the evaluation of all SPR experiments. This also applies to monomeric compounds as well as Homo-BacPROTACs, which contain two binding moieties per molecule.

For the stacking experiment, 468 RU of ClpC1-NTD were immobilized on a CM5 chip as described above. The experiment was performed in dual injection mode on an 8k instrument. In short, in the first injection, Homo-BacPROTAC was injected at a fixed concentration of 200 nM. Subsequently, two different concentrations of ClpC1-NTD were injected.

### Cell-free degradation assay

The assay was adapted from Morreale et al.[20], with the following modifications:

The assay was performed with degradation machineries from *M. smegmatis* and *M. tuberculosis*.

The final assay mixture contained 0.5 μM *M. smegmatis* or *M. tuberculosis* ClpC1 (hexameric), 0.25 μM *M. smegmatis* or *M. tuberculosis* ClpP1P2 (tetradecameric), 1.5 μM substrate protein *M. tuberculosis* ClpC1NTD, 15 mM phosphoenolpyruvate (PEP, Sigma Aldrich #10108294001), 10 U/ml pyruvate kinase (Sigma Aldrich, #P7768 or P9136). It was performed in a buffer containing 50 mM HEPES pH 7.0, 100 mM KCl, 40 mM MgCl₂, 10% (v/v) glycerol.

Compounds were dissolved in 100% (v/v) DMSO, pre-diluted 1:10 with buffer to 10x assay concentration. The final concentration of DMSO in the assay was 1% (v/v). In control experiments, DMSO was added to the same concentration. Reactions were performed in 96-well plates (Eppendorf #951020303, final reaction volume 8 μl). Only the inner 60 wells of each plate were used and wells adjacent to the

reaction wells were filled with a matching volume of water. All components except for the compounds and ATP were combined in a master mix. After addition of the compounds to the pre-mix, degradation was started by addition of ATP to a final concentration of 5 mM. The reaction plate was sealed and incubated at 37 °C in an Eppendorf Thermomixer C with heated lid (400 rpm, 90 min) During incubation, a second plate was prepared containing 1.5 μl 5x Fluorescent Master Mix (ProteinSimple) per well, spun down and stored at room temperature. At the end of the incubation, the assay plate was transferred to ice for 30 s and then centrifuged for 5 min at 2200 × g. Reactions were subsequently diluted by adding 2 μl of 0.1 × sample buffer (ProteinSimple) to each well. Next, 6 μl of each reaction were added to the Fluorescent Master Mix, the plate was sealed and incubated at 95 °C for 5 min (non-heated lid). The plates were then allowed to cool down to room temperature and finally centrifuged for 5 min at 2200 x g before analysis.

The WES capillary-based immunoassay platform (ProteinSimple) was used for measurement of the samples according to the manufacturer's instructions. Anti-His tag (0.5 μg/ml, R&D Systems MAB050, for ClpC1 NTD detection) and 6x His tag (Abcam, ab252883, 0.02 μg/ml and 0.1 μg/ml, or ab206500, 0.05 μg/ml and 0.1 μg/ml, for *M. smegmatis* and *M. tuberculosis* ClpP1P2 detection, respectively) were used as primary antibodies, incubated for 60 min and detected using the anti-mouse detection module (ProteinSimple DM-002).

Data analysis was performed using Compass for SW Software (ProteinSimple). Signals were quantified and peaks named (ClpC1 NTD 20–21 kDa, ClpP1P2 26–27 kDa) and the following peak fitting settings were used: Peak Find Threshold 100.0 and Area Calculation by Dropped Lines. All other settings used were the default values.

Resulting ClpC1 NTD areas were divided by ClpP1P2 areas and normalized to the negative control (DMSO/no compound).

$DC_{50}$ values were calculated from these values with Boehringer Ingelheim's MEGALAB $DC_{50}$ application using a 4 parametric logistic

model. The value of the lower asymptote was constrained to >0 % for the calculation of the $DC_{50}$ value. The upper asymptote was constrained to <130%. At least three independent measurements with the *M. smegmatis* machinery and at least two with the *M. tuberculosis* machinery were performed for each compound with mean values and standard deviations reported. All experiments were performed in triplicates. These well replicates were prepared by treating three reactions from the same master mix in the same plate with the same compound dilution.

For SDS-PAGE experiments, reaction samples were treated with 2.7 µl 4x Roti Load 1 (Roth #K929.1), instead of adding sample buffer and fluorescent mastermix, and incubated at 95 °C for 5 min. Samples were then loaded on a 4–12% Bis-Tris Criterion XT Precast gel (BIO-RAD #3450125). The PageRuler Prestained Protein Ladder (Thermo Scientific #26616) was used as a marker. Gels were run in BIO-RAD Criterion cells with a PowerPac Basic (BIO-RAD) and stained using SYPRO™ Ruby Protein Gel Stain (Invitrogen S12000) following the protocol provided by the manufacturer with a few modifications: Gels were incubated in 2 × 100 ml fix solution (50% (v/v) methanol, 7% (v/v) acetic acid) for 30 min, then 100 ml SYPRO™ Ruby protein gel stain overnight and finally in 200 ml wash solution (10% (v/v) methanol, 7% (v/v) acetic acid) for 60 min. After washing the gel in water for 3 × 5 min, it was imaged in an Amersham Imager 680 (Cytiva) on a white trans tray, using a Cy3 detection filter (Epi-RGB = Filter Green 520 nm) and an exposure time of 2 s.

To monitor degradation over time, individual reactions á 8 µl per timepoint and replicate were incubated in 1.5 ml Eppendorf tubes at 37 °C and 400 rpm (Eppendorf Thermomixer compact/comfort). Reactions were stopped after 0, 45, 90, 150, 240 and 360 min. To stop the reactions, tubes were transferred to ice for 10 s and spun down briefly, followed by the addition of 2 µl 0.1x sample buffer to each tube and subsequent treatment with 5x Fluorescent Master Mix as described above. Samples were measured using the WES platform as described. Degradation experiments over time were performed in duplicates, which were prepared by treating two reactions from the same master mix with the same compound dilution. DMSO served as a negative control. Compound 8 was tested at 8 µM and 33 µM.

Experiments were analyzed in Compass for SW software as described above. For further analysis, peak areas were exported to MS Excel. ClpC1 NTD peak areas were divided by ClpP1P2 peak areas and treated samples were normalized against the according DMSO controls for each timepoint. Data were visualized using GraphPad Prism 9.5.0.

## Intracellular degradation assay

The intracellular ClpC1 degradation assay was adapted from Morreale et al.[20] as follows:

Compounds dissolved in DMSO were serially diluted by 8-fold to 100x final assay concentration, covering a range of 0.0004–100 µM, and 1 µl compound solution per well was added to a glass-coated 96-well plate (Thermo Scientific #60180-P330). For the negative control, only DMSO was added. Each experiment was performed in triplicates. These well replicates were prepared by adding the same compound dilution to three different wells, each of which would later be filled with the same cell suspension.

*M. smegmatis* (ATCC™ 607™) cultures were grown in Middlebrook 7H9 liquid culture medium (Sigma #M0178, 0.025% (v/v) Tween80, 0.1% (v/v) glycerol) at 37 °C and 180 rpm. An overnight culture grown to an $OD_{600}$ = 0.2–0.3 was centrifuged at 2264 × *g*, 23 °C, 5 min and concentrated by factor 5. Per well, 100 µl concentrated cell suspension were added, mixed, and sealed. The plate was incubated for 24 h at room temperature without agitation.

To harvest the cells, 90 µl cell suspension from each well were centrifuged at 4000 × *g* and 23 °C for 3 min. The cell pellet was resuspended in 100 µl cold lysis buffer (50 mM HEPES pH 7.5, 150 mM

KCl, 10% (v/v) glycerol) containing complete protease inhibitor cocktail (EDTA-free, Roche) and stored on ice until lysis.

Cells were lysed in a Bioruptor Pico in 10 cycles, 30 s on, 30 s off, at 4 °C and then centrifuged at 21000 × *g* and 4 °C for 30 min. The supernatant was flash-frozen in liquid nitrogen and stored at −80 °C.

Lysates were analyzed by JESS Total Protein Normalization (TPN, ProteinSimple) using the Replex Module (ProteinSimple #RP-001) and the Total Protein Detection Module (ProteinSimple #DM-TP01). Samples were treated and plates prepared according to the manufacturer's instructions. For the detection of ClpC1, anti-ClpC1 C-terminus antibody from sheep at a concentration of 0.005 mg/ml and anti-sheep secondary HRP-conjugated antibody 1:50 (R&D anti-sheep IgG #HAF016) were used. Antibodies used for detection of ClpC1 from *M. smegmatis* and *M. tuberculosis* were generated at MRCPPU Reagents and Services, University of Dundee, UK, by immunizing sheep with the following peptides: MFERFTDRARRVVVLAQEEAR (derived from the N-terminus of ClpC1, corresponding to amino acids 1–21, 100% conserved between *M. smegmatis* and *M. tuberculosis*) and RRTIQ-REIEDQLSEKILFEEV (derived from the C-terminus of ClpC1, corresponding to amino acids 774–794, 100% conserved between *M. smegmatis* and *M. tuberculosis*). All in vivo work was performed in the United Kingdom under ethical approval and UK Government Home Office licence authority. The standard JESS TPN protocol was used with 60 min primary antibody incubation.

Jess TPN data were analyzed with Compass for SW Software: The ClpC1 peak was fitted with Peak Find Threshold set to 100.0. Resulting ClpC1 peak areas were divided by total protein areas and normalized to the negative control (DMSO/no compound).

From these values, $DC_{50}$ values were calculated with Boehringer Ingelheim's MEGALAB $DC_{50}$ application using a four-parametric logistic model. For the calculation of the $DC_{50}$ value, the values of the lower and upper asymptote were constrained to >0 % and <130 %, respectively. Three or four independent measurements were performed for each compound with mean values reported.

## Chemicals and media used for microbiology

The reference drugs or antibiotics viz. Isoniazid (INH), Rifampicin (RIF) and Moxifloxacin (MOX) were procured from Merck USA (erstwhile Sigma-Aldrich). Stock solutions of different drugs were prepared in the respective recommended solvents (e.g. dimethyl sulfoxide (DMSO) for RIF and MOX, or in Milli-Q water for INH). The working solutions were prepared fresh every time at the beginning of the experiment. The various media used for growing different strains at FNDR were: 1) mycobacteria were grown in Middlebrook 7H9 broth supplemented with 10% (v/v) Middlebrook ADC, 0.05% (v/v) Tween-80 and 0.25% (v/v) glycerol), whereas the Gram-positive and Gram-negative strains from ESKAPES panel were grown in Mueller Hinton media (BD/Difco). All the bacteria used at FNDR were grown to a cell number of $10^9$ colony-forming units (CFU)/mL and were preserved as glycerol-stocks at −80 °C in 0.5 ml aliquots. A single vial was thawed and used every time for each experiment.

The *Mtb* Beijing lineage HN878 was obtained from the laboratory of Dr. William R. Jacobs Jr., Albert Einstein College of Medicine, Bronx, NY and was grown at the Institute for Pharmaceutical Biology and Biotechnology, Heinrich-Heine-University Düsseldorf, Germany at 37 °C and 80 rpm in liquid Middlebrook 7H9 medium supplemented with 0.2% glucose, 0.085% sodium chloride, 0.5% glycerol and 0.05% tyloxapol.

## MIC resazurin microtiter assay for *Mycobacterium tuberculosis* and Nontuberculous Mycobacteria (NTM)

Except where stated otherwise, minimum inhibitory concentrations (MICs) were determined at the Foundation for Neglected Disease Research (FNDR) against the mycobacterial strains by the standard broth dilution method according to Clinical Laboratory Standards

Institute[38,39] guidelines M24. The different mycobacterial strains used were *M. smegmatis* (*Msm*), *M. abscessus* (*Mabs*), *M. fortuitum* (*Mfo*), *M. avium* (*Mav*), *M. intracellulare* (*Mint*), and different *M. tuberculosis* (*Mtb*) strains. The different *Mtb* strains used were drug-sensitive strains (H37Rv ATCC 27294 and 11291), mono drug resistant strains Isoniazid resistant H37Rv (katG[del]) ATCC 35822[40], Rifampicin resistant H37Rv (rpoB[S450L]) ATCC 35838[41], moxifloxacin resistant H37Rv clinical isolate FNDR-M1 as well as multidrug resistant strains (ATCC 35825 and 8673) with drug-resistance against 1st and 2nd-line anti-TB drugs. Briefly, the test compounds were dissolved in DMSO, serially diluted by 2-fold in a 10-concentration dose response (10c-DR) ranging from 128 to 0.25 μg ml⁻¹ in 96-well plates. Middlebrook 7H9 broth (containing 10% albumin dextrose catalase supplement, ADC) complete media was used for the assay. Each mycobacterial culture was added as 200 μl in each well to all columns except the media control column (200 μl of media was added) to give a final inoculum of $3–7 \times 10^5$ cfu/ml.

The quality control (QC) included: media controls, growth controls (including DMSO controls), and the assay specific appropriate reference compounds Rifampicin, Isoniazid, Moxifloxacin, as well as the MDR-*Mtb* specific reference drug controls: Amikacin, Thiacetazone and p-Aminosalicylic acid. The assay plates were incubated at 37 °C, resazurin dye was added on the 3rd day for *Msm, Mabs, Mfo*; and on the 6th day for *Mav, Mint*, as well as for the sensitive or MDR *Mtb* strains.

The results were noted on the 4th day for *Msm, Mabs, Mfo* and on the 7th day for *Mtb, Mav, Mint* using colorimetric readout. The MIC of the compounds and the reference drugs were recorded. The blue wells indicated inhibition of growth, while the pink wells indicated uninhibited growth. MIC assays were carried out in duplicates or triplicates. MIC is defined as the minimum concentration of any compound that inhibits mycobacterial growth or prevents the colour change from blue to pink at the end of the respective assay period.

For antibacterial activity testing against replicating cells of *Mtb* Beijing lineage HN878, cultures obtained from exponentially growing bacteria were adjusted and seeded at $1 \times 10^5$ CFU/well in 96-well round bottom microplates, in a total volume of 100 μl containing two-fold serially diluted test compounds with a starting concentration of 100 μM. The plates were incubated as standing cultures at 37 °C for 5 days. 10 μl of a 100 μg ml⁻¹ resazurin solution was subsequently added to each well and further incubated for 16 h. To fix the cells, 100 μl of 10% formalin was added to each well (5% final concentration) and incubated for at least 30 min. Subsequently, fluorescence was quantified using a microplate reader (Tecan) (excitation: 540 nm; emission: 590 nm). Percentage of growth was calculated relative to sterile medium (0% growth) and solvent control (100% growth).

### Starvation-induced non-replicating persistence model
These assays were performed at the Institute for Pharmaceutical Biology and Biotechnology, Heinrich-Heine-University Düsseldorf, Germany. To test the activity of compounds against non-replicating cells of *Mtb* H37Rv, cells were grown to stationary phase, harvested, washed thrice with PBS-0.025% tyloxapol, resuspended in PBS-0.025% tyloxapol in the original culture volume and starved by incubation at 37 °C for three weeks. Next, cells were diluted to $1 \times 10^8$ CFU ml⁻¹ with PBS-0.025% tyloxapol and transferred into 96-well round bottom microtiter plates to a final volume of 100 μl per well, and compounds were added at the indicated final concentrations. After five days of incubation at 37 °C as standing cultures, resazurin solution (10 μl/well from 100 μg ml⁻¹ stock) was added, and cells were incubated for 48 h at 37 °C. Subsequently, cells were fixed and fluorescence was measured as described above.

### Determination of Minimum Bactericidal Concentration (MBC)
The minimum bactericidal concentration (MBC) of the respective compounds against *Mycobacterium tuberculosis* (Mtb) or *Mycobacterium smegmatis* (Msm) were evaluated at the FNDR by processing the

culture aliquots from the parallel MIC assay plates. Culture aliquots of 100 μl volume from all wells showing growth inhibition were subcultured by spread plating onto compound-free Middlebrook 7H9 agar plates supplemented with 10% ADC. These plates were incubated at 37 °C with 5% $CO_2$ for 4 days for *Msm* and 4 weeks for *Mtb*. The colony forming units per ml (CFU/ml) were enumerated and recorded. MBC was defined as the lowest concentration of the compound that killed or reduced the initial mycobacterial load by >100 fold.

### Minimum Inhibitory Concentration (MIC) assay against ESKAPES panel
Minimum inhibitory concentration (MIC) against ESKAPES panel of Gram positive and Gram negative pathogens (*Enterococcus faecium, Staphylococcus aureus, Klebsiella pneumoniae, Acinetobacter baumannii, Pseudomonas aeruginosa, Enterobacter cloacae* and *Salmonella typhimurium* [ESKAPES panel]) were determined at the FNDR by the standard broth dilution method [CLSI, M100]. Briefly, the test compounds were dissolved in DMSO, serially double-diluted in a 10-concentration dose response ranging from 64–0.125 μg/ml in 96-well plates. Mueller Hinton broth (MHB) medium was used to grow ESKAPES panel strains. The individual cultures were added as 200 μL (inoculum of $3–7 \times 10^5$ cfu/ml) to the respective assay plates in each well to all columns, except the media control (200 μL of medium). Quality controls included: media controls, growth controls, and reference drug inhibitors (Moxifloxacin or any other assay-specific reference drug/s). The assay plates were incubated at 37 °C, and the results were noted on the 2nd day as turbidometric readout. The clear wells indicated inhibition of growth, while the turbid wells indicated uninhibited growth.

### Intracellular MIC assay in THP-1 macrophages
These experiments were conducted at the FNDR. To test drug efficacy against *M. tuberculosis* H37Rv in the intracellular compartment, the monocytic macrophage cell line THP-1 (ATCC TIB-202) was used. The THP-1 cells were grown in RPMI medium (Gibco-BRL Life Technologies, Gaithersburg, Md.) in 75 cm² flasks at 37 °C in a 5% $CO_2$ atmosphere. RPMI media was supplemented with 100 mM sodium pyruvate, 200 mM L-glutamine, 3.7 g of sodium bicarbonate per liter and 10% fetal calf serum without any antibiotics. After counting the THP-1 cells in a hemocytometer, viability was determined by trypan blue exclusion, and the cells were seeded in 96-well plates in duplicates with complete RPMI containing 10% fetal calf serum at a density of $1 \times 10^5$ cells/well ($5 \times 10^5$ cells/ml) and were incubated overnight.

Differentiation of THP-1 cells was induced by incubation of THP-1 cells with 50 nM phorbol 12-myristate 13-acetate (PMA) for 72 h. After this induction, the THP-1 macrophages were infected with *M. tuberculosis* H37Rv strain (ATCC-27294) at a multiplicity of infection (MOI) of 1:10 in the fresh media. The macrophage infection was allowed to take place for 2 h at 37 °C with 5% $CO_2$. The media containing *M. tuberculosis* was discarded, macrophage monolayers were washed twice with phosphate-buffered saline (with $Ca^{2+}$ and $Mg^{2+}$) to remove the free bacteria and replenished with the fresh complete RPMI.

For the day-0 infection control, sets of duplicate wells were lysed (0.05% SDS) and enumerated to assess the numbers of intracellular *M. tuberculosis* 2 h post-infection at the beginning of treatment.

For the remaining wells, at 2 h post-infection, the treatment was initiated (test and reference compounds in DMSO) in comparison with the respective no-treatment control (DMSO only). The test and the reference (moxifloxacin and rifampicin) compounds were added to the sets of duplicate wells at indicated concentrations. The final concentration of DMSO in the medium was maintained at 1% for all conditions including the no-drug infection control. Sets of replicates from the infection control, test and reference wells with each drug concentrations were sampled on different time points (days 4 and 7). The wells at the respective time points were washed to remove the

extracellular bacteria. The monolayers lysed with 0.05% SDS were serially diluted and plated onto Middlebrook 7H11 agar plates to enumerate the number of intracellular viable mycobacteria as colony-forming units (CFU/ml). The intracellular mycobacterial killing rates for each concentration of test and reference compounds were generated by plotting the $\log_{10}$ cfu/ml against 0, 4, and 7 days. Maximal antimicrobial effect (Emax) was determined by the formula: Start $\log_{10}$ cfu/ml (D-0)–Residual $\log_{10}$ cfu/ml (D-4 or D-7).

### Microsomal stability assays

The kinetics of compound degradation in liver microsomes is assessed in an assay with 1 µmol/l of compound and 0.5 mg/ml liver microsomes in a medium of 100 mM Tris-HCl pH 7.5, 6.5 mM $MgCl_2$ and 1 mM NADPH at 37 °C. The reaction is stopped at multiple time points by adding acetonitrile. After centrifugation, the compound concentration is measured in the supernatants by high-performance liquid chromatography (HPLC) tandem mass spectrometry (MS/MS). The results are fitted to a first-order decay and the half-lives are converted to clearance (stated as % of the liver blood flow (%Qh)) using the well-stirred liver model.

### Hepatocyte stability assays

Cryopreserved hepatocytes are incubated in an appropriate buffer system containing 50% species serum. Following an acclimation period (15–30 min) in an incubator (37 °C, 5–10% $CO_2$, 85–95% humidity) the test compound is added to the hepatocyte suspension (pH 7.4, typical cell density of about 1 million cells/mL; final concentration of test compound is 1 µM, final DMSO concentration <0.05% v/v). The cells are incubated for up to 6 h and samples are taken at 6 different time points. Samples are then quenched with acetonitrile and pelleted by centrifugation. The remaining amount of parent compound in the supernatants is then analysed by HPLC-MS/MS. Clearance is calculated from compound half-lives using the well-stirred liver model and is converted to clearance stated as % of the liver blood flow (% Qh).

### PK following iv bolus and po administration to BALB/cAnNCrl mice

**Health status and animal husbandry.** The in-life experimental procedures were conducted in strict compliance with German and European animal welfare legislation in an AAALAC-accredited facility. Male BALB/cAnNCrl mice (Charles River Laboratories Research Models and Services, 97633 Sulzfeld, Germany) were used. Animals were 9 weeks old at dosing with a body weight of 21.6–24.3 g. Animals were delivered free of pathogens according to FELASA recommendations with a health certificate provided by the breeder. Upon receipt, the state of health and sex was checked. Animals acclimatized for at least 5 days before the commencement of the in-life phase. During acclimation, animals were group-housed in individually ventilated cages, on birch wood granulate bedding with species-specific enrichment (nesting material, mouse igloo, gnawing wood). During experimental conduct, animals were housed in conventional cages with elevated grid floors. Grid floor was used to avoid contamination with test item material excreted via urine and feces and subsequent reuptake during fur grooming. No bedding and enrichment were provided except a cage divider, which can be used as a shelter. Animals always had unlimited access to food and water and received a pelleted, total pathogen-free maintenance diet and tap drinking water, sterilized by filter.

**Formulation (iv).** On dosing day, test item(s) was dissolved in DMSO (2.86 mg/ml), 13.3 vol. of PEG200 (40% in water for injection) added, and sonicated at 60 °C for 10 min. In case of incomplete dissolution, the formulation was filtered at 40 °C (non-pyrogenic sterile-R filter, 0.2 µm). All formulations were stirred warm until administration. Actual test item content was determined bioanalytically.

**Formulation (po).** On the day before dosing, test item(s) were suspended in kleptose (20% in WFI) under continuous stirring (3.0 mg/ml). Following 2 min vortexing and 1 h stirring, the formulation was sonicated at 60 °C for 30 min and stirred at ambient temperature overnight. Before dosing, the formulation was vortexed for 2 min followed by ultraturrax treatment for 2 min.

**In life phase.** Male BALB/cAnNCrl mice (n = 3 per test item and administration route) received a bolus dose of the formulated test item(s) (1 mg/kg iv, 30 mg/kg po) at t = 0 h, and whole blood samples were collected subsequently (35 µl at 0.1 (iv) / 0.25 (po), 0.5, 1, 2, 4, 6 h; 600 µl at sacrifice after 24 h). Animals were euthanized by exsanguination in deep isoflurane anesthesia. Whole blood samples were immediately stored on ice at about 4 °C. Plasma was separated by centrifugation at approximately 10,000 × g for 5 m at +4 °C. Centrifugation of each blood sample was started within 10 m after collection. A single aliquot of at least 15 µL of plasma was prepared and immediately stored at −20 ± 5 °C.

**Bioanalytics.** Sample proteins were precipitated with ACN (1 + 12, v/v) containing internal standard, and supernatants analyzed by quantitative UPLC-MS/MS: Phenomenex Luna Omega C18, 2.1 × 50 mm, 1.6 µm, 40 °C, gradient between 10% - 90% ACN with 0.1% formic acid. LogD determination: Lipophilicity of the test items, expressed as chromLogD$_{7.5}$, was assessed by measuring their chromatographic retention times on C18-material. The chromatographic retention versus logD$_{7.5}$ values of a series of thirteen n-alkan-2-ones (2-butanone – 2-hexadecanone) served as lipophilicity calibration. Chromatographic conditions were as follows: HPLC-UV Agilent 1260/1290 system, UV-detection: 270 nm, $C_{18}$-column, flowrate: 1.3 ml/min, run time: 6 min, injection volume: 2.0 µl, eluents: A: $NH_4OAc$-buffer pH 7.5 and B: ACN, Gradient: 0 min: 90% A – 2.60 min: 5% A – 3.55 min: 90% A.

### Chemical synthesis

Full details of synthetic procedures and NMR spectra of all compounds (Supplementary Figs. 8–209) are provided in the supplementary information.

All further methods and all data supporting the findings of this study are available within the Supplementary Information.

### Reporting summary

Further information on research design is available in the Nature Portfolio Reporting Summary linked to this article.

## Data availability

The authors declare that the data supporting the findings of this study are available within the paper, the Source Data file and the Supplementary Information. The SPR data generated in this study are provided in the Source Data file as a separate ZIP folder. The NMR data generated in this study are provided in the Supplementary Information. Should any raw data files be needed in another format they are available from the corresponding authors upon request. Source data are provided in this paper.

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

## Acknowledgements

We are grateful to Dieter Gallemann for providing advice on DMPK profiling and coordinating the work packages at Nuvisan GmbH. We thank James Hastie and the team at MRCPPU Reagents and Services, University of Dundee, United Kingdom for the generation of ClpC1-specific antibodies. Maximilian Scharnweber is acknowledged for valuable input in discussions of biophysical data. Editing support was provided by Life Science Editors. This work was supported by the European Research Council (AdG 694978, T.C.), an FFG Headquarter Grant (no. 852936, T.C.), the Vienna Science and Technology Fund (WWTF LS17-029, T.C.) and Boehringer Ingelheim (L. J., S. G., J. L). Financial support from Saarland University and the DFG (grant 447298507) is gratefully acknowledged. R.K. acknowledges support from the DFG (project number 270650915/GRK 2158) and the Jürgen Manchot Stiftung (graduate school MOI IV). J. L., F. E. M and S. J. are members of the Boehringer Ingelheim Discovery Research global post-doc program. The IMP is supported by Boehringer Ingelheim.

## Author contributions

L. J., V. M. S., K. R., P. K., S. N., R.K.S., A. Meinhart, T. C., R.K. and G. B. designed experiments. L. J., S. G., P. B., and G. G. performed the chemical synthesis and analysis of the compounds. P.K., R. V. K., J. L., S. J., K. V., V.K. and L.v.G. performed mycobacterial assays including those under BSL3 level. K. F., J. L., D. H., S. J. and F. E. M. performed microbiological and biochemical assays, as well as binding measurements including analyses. K. R. designed and supervised SPR experiments. P. G. coordinated the synthesis and supply of compound building blocks.

L. J., V. M. S., P. K., U. K., T. C., M.S., H. W. and G. B. coordinated the research collaborations between Boehringer Ingelheim, Saarland University, IMP and FNDR. L. J., V. M. S., C. K., A. Mantoulidis and H. W. contributed to the design of reported compounds. L. J., V. M. S., H. W. and G.B. prepared the manuscript together with input from all authors.

## Competing interests

V. M. S., C. K., A. Mantoulidis, P. G., H. W., K. R., K. F., P. B., G. G. and G. B. were employees of Boehringer Ingelheim at the time of this work. The remaining authors declare no competing interests.

### Ethical compliance

The authors confirm that the research in this study complies with all relevant ethical regulations. All animal studies were approved by the District Government of Upper Bavaria (Regierung von Oberbayern, Az.: 55.2-2532.Vet_03-17-101). All in-life experimental procedures were conducted in strict accordance with the protocol and in compliance with German and European animal welfare legislation.
