## [Peer Review File · Nature Communications]

REVIEWER COMMENTS

Reviewer #1 (Remarks to the Author):

Junk et al., in their manuscript entitled “BacPROTAC-induced degradation of ClpC1 as a new strategy against drug-resistant mycobacteria,” demonstrated for the first time the targeted protein degradation of an endogenous substrate in mycobacteria by the bacterial protease machinery, ClpC1P1P2.

This work is impactful because it shows homo dimeric binders of ClpC1, which is an essential component of the bacterial proteolytic machinery ClpC1P1P2, can also act as degraders to induce its own degradation. Previously, the original paper on the discovery of BacPROTACs by Morreale et al. (Cell 2022) provided the proof of concept that BacPROTACs can recruit the Clp-mediated degradation of non-endogenous model proteins of interest and fusion proteins in mycobacteria. In this manuscript, Junk et al. build on this impactful scientific premise to show how cyclomarins-based cyclic heptapeptides, possessing anti-Mtb-activity, with two ligands of ClpC1 direct the Clp protease machinery against its own ClpC1 subunit resulting in self-destruction.

This work has been thoroughly conducted, supporting its claims that homodimeric BacPROTACs exhibit improved anti-Mtb activity. Experiments supporting the claims include drug-resistant strains and in macrophages through experiments through degradation assays of ClpC1NTD comparing with ClpC1P2 and full-length ClpC1 both in cell-free assay systems and intracellular systems. Assessing MICs and MBCs against Msm #607, Mtb H37Rv strains to showcase these BacPROTACs not inhibiting gram +ve, gram -ve bacteria, or other NTMs.

Due to the noteworthy scientific findings, the manuscript is recommended for publication, despite some limitations in the physicochemical and PK properties of the BacPROTACs associated with solubility and bioavailability. The suggested minor revisions listed below would make the claims more robust or enable the reproducibility of the experimental data provided in this paper.

1. The following experimental data would bolster the claims against the activity of the BacPROTACs against drug-resistant strains of Mtb.

- Performing the assays against RIF-resistant strains (H37Rv rpoBS450L) since RIF is one of four drugs used in the first-line treatment of TB
- Standard isoniazid-resistant (INH: katGdel) strain (MS015), and a moxifloxacin-resistant (MOX: gyrAD94K) strains
- Hypervirulent HN878 Mtb strain of the W/Beijing Lineage (Lineage 2), which is increasing in incidence in active TB cases and is frequently associated with the occurrence of drug resistance
- Listing the MIC values of the TB drugs, Streptomycin, RIF, INH, etc., in Table 3 instead of stating sensitive and resistant.

2. Decimal points and commas are interchangeably used as decimal markers. One type of decimal marker needs to be used as a marker.

3. In the SI, listing the diagnostic peaks in ^1H and ^{13}C NMR will be helpful for the readers in distinguishing major versus minor rotamer.
4. In the SI ^{13}C NMR, second decimal digits need to be listed to distinguish identical peaks; e.g., page 21 compound 7 has two peaks at 171.6 ppm.
5. The literature reference needs to be cited for the known compounds in the intermediates for the enantiomeric BacPROTACs. If they are new compounds, full characterization data need to be provided.
6. Many spectra contain impurities and solvent peaks for ethyl acetate, acetone, and dichloromethane. Cleaner spectra need to be provided for the compounds SI-7, 8, 9, 10, 15, 19, 21, 25, 26, 28, 30, 31, 36, 37 (poor resolution for ^{13}C NMR), 47.

Reviewer #2 (Remarks to the Author):

Cyclomarins (Cym), cyclic peptides produced by certain marine *Streptomyces* sp, are known to have antitubercular activity. They were shown to exert their toxicity by binding to the essential protease ClpC1. In their recent study, Junk et al redesign Cym to not simply bind the target ClpC1 but instead cause its self-mediated degradation. The strategy is based on synthesis of a crosslinked Cym dimer. These compounds possess two ClpC1 binding entities such that ClpC1 protomers can be bound at one end and the ClpC1P degradation machine at the other, leading to proximity-induced degradation of the ClpC1 protomer by the ClpC1P complex.

The authors achieve this by introducing an alkyne moiety into the Cym molecule at specific positions, and then use a bifunctional crosslinker carrying two azide groups to ligate the two Cym molecules using azide-alkyne cycloaddition. The authors test different Cym derivatives with the aim to maximize eukaryotic cell permeability and biological activity. They can demonstrate that dimeric Cym outperform their monomeric counterparts in their effect on *Mtb*, hence providing more potent compounds that address ClpC1 as a drug target.

The study is interesting and timely, and the protease self-targeting approach is novel. I support publication.

However, several points should be addressed prior to publication:

- 1) PROTACS are usually bifunctional binders that feature one moiety that binds the target protein to be degraded and another moiety that binds the protease (or a ligase that tags the target protein for degradation). The Cym crosslinked dimers are not bifunctional, hence the term homo-BacPROTAC, since they serve the very specific task to target the protease for self-degradation. As such, Cym dimers do not represent a general strategy that could render any mycobacterial protein a degradation substrate, but they are ClpC1-specific drugs. Overstatements concerning the applicability should be avoided. The title should reflect the self-targeting nature of the approach. For example: "Homo-BacPROTAC-induced ClpC1-mediated self-degradation as a new strategy...".
- 2) In vitro degradation assays: The reaction mixture was incubated for 90 min for each Cym concentration. Why was this time frame chosen? An actual time course should be shown for BacPROTAC 8 at the DC50 and at the highest tested concentration. Does it take 90 minutes to achieve degradation of

half the ClpC1 N-terminal domain present, and when further time points are taken at 120 min and later, does the ClpC1 N-terminal domain eventually disappear completely or not?

3) The effect of binding-competent Cym monomers (5 and dCymC) on ClpC1 in vivo levels should also be tested and included in Figure 5 to show that the degradation observed in vivo is mediated only by the dimeric Cyms and not the monomeric Cyms.

4) The core result of the study is the strongly improved MIC compared to smaller changes in the binding of ClpC1 for the dimeric versus monomeric Cyms, highlighting the power of harnessing an enzyme instead of relying on binding. It is therefore concerning that this appears to only hold true for Mtb and not for any of the other mycobacteria. For *M. smegmatis* the MICs for the dimeric and monomeric Cyms were comparable and for other pathogenic mycobacteria, only the monomeric Cyms showed growth inhibition.

Given the high homology between mycobacterial ClpC1, how can this be explained? Have the authors assessed the conservation of the known Cym binding residues in ClpC1 orthologs? Furthermore, if monomeric Cyms bind, why would the dimeric Cym not work? The authors should test the permeability of monomeric and dimeric Cyms into the different mycobacterial organisms to assess if their cell walls present different barriers.

Minor comment:

The labels in Figure 4 are too small.

Reviewer #3 (Remarks to the Author):

This manuscript is very interesting to show Homo-BacPROTACs for degrading ClpC1. I have some concerns:

1. The binding affinity is very high for the dimer in low nanomolar range even picomolar, however, the degradation is much lower reflecting by the DC50 and Dmax in high micromolar. Whereas, inhibition of the Mtb growth is much better for 12 (97 nM). My concern is: whether the degradation is predominated for the biological activity? Or other mechanism in the anti-bacteria activity?

2. As the author mentioned, homo-PROTAC has several limitations, e.g. physicochemical properties, solubility, polarity, bioavailability Therefore, what are the strengths for the homo-PROTAC? Only a new concept? How about the in vivo efficacy? What is the difference for homo-PROTAC compared with biological method?

Reviewer #4 (Remarks to the Author):

This important manuscript examines Bac-PROTACs as a new strategy for treating drug-resistant tuberculosis. Targeted protein degradation is an important emerging technology, which has thus far primarily been applied to Eukaryotic systems by manipulating the ubiquitin-proteasome degradation system to destroy rather than merely inhibit therapeutic targets. In this manuscript, ClpC1 targeting BacPROTACs are generated and examined starting from cyclomarin, a natural product that binds ClpC1

NTD. The most potent compounds produced (homodimeric-cyclomarin analogs) show potent anti-tuberculosis activity including against multi-drug resistant strains, an exciting result.

This chemistry-focused manuscript is highly rigorous, with extensive biological and chemical data presented in the main manuscript, in the supplementary data section, and in an associated manuscript under review, giving high confidence in the reliability and reproducibility of the results presented. The chemistry design for the series generation is well rationalized and easy to follow, with the consideration of structural changes to increase the permeability and pharmacological properties of the series particularly well conveyed.

The primary weakness of this manuscript is that there is not adequate validation that the ClpC1 NTD is the target of the Bac-PROTACs 8 and 12, within the cell. Proteomic experiments in the associated manuscript suggest that this is true, but the routine methods for demonstrating on-target inhibition in Mtb through resistant mutant generation and sequencing, or ClpC1 binding site mutagenesis, have not been explored. This is a concern, as the introduction highlights the importance of antimicrobial resistance and the need for new therapeutic approaches, yet the propensity for resistance development for this new therapeutic class is not adequately considered.

In summary, some additional MOA evidence is needed in this otherwise exceptional manuscript, which is judged as highly suitable for Nature Communications.

Further suggestions:

1. The anti-tubercular activity of 8 and 12 should also be examined against non-growing Mtb cells as this is a major therapeutic need.
2. Line 112 – “we show” or if we showed provide the citation
3. Line 195 - Why Caco-2 and not PAMPA when considering cellular permeability, these molecules are highly unlikely to be oral agents?
4. Line 214 – The statement is incorrect the vectors have lower AUCs and higher clearance than the corresponding BacPROTACs
5. Line 252 – Please rationalize why the BacPROTACs degradation is at a much higher concentration than the Mtb MIC – does this suggest there is another target?
6. Figure 6 graphs are very hard to read
7. SPR sensorgrams should be provided in full in the supplementary data section for all key molecules
8. Table 3 MIC values for the control antibiotics (Strep, INH< Rif, Ami,) determined at the same time as homo-bacPROTACs should be reported as numerical values
9. Control compound values are missing from the supplementary data table S1 for instance high/medium/low Caco-2 controls – a moderate concern
10. Raw data for the IV PK testing should be provided in either a data table or graphic form
11. S15 General information (chemistry), please provide a statement on how compound purity was determined

Reviewer #5 (Remarks to the Author):

Key results: The authors present the synthesis of homodimeric forms of cyclomarins and demonstrate that these compounds may lead to the degradation of ClpC1 and antibacterial activity.

Significance to the field: As a nonexpert in the field of antimicrobials it is difficult to judge the significance and novelty of the molecules themselves which were previously known to bind ClpC1 as monomers.

The editors requested input on the application of SPR. It is curious to see the authors present 200 pages of supplemental data which include 196 figures of NMR data sets but not a single SPR data set. The supplemental should include figures of all the SPR data overlaid with fits to the model used to extract the binding constants.

In order for others to be able to reproduce the SPR assays, the amount of target captured for each study should also be included with the figure along with a list of the analyte concentrations that were used in each injection.

The authors should also clearly state what they used for the molar concentrations of the dimeric compounds compared to the monomeric forms.

The authors claim the increase in the binding affinity for the homodimeric forms of dCym compared to monomeric forms is not due to avidity. However, the method the authors report using (comparing KD vs surface density) is not conclusive. Since they are using a CM5 surface (which is coated with a flexible dextran layer) and they are limited in how low they can go in surface density (because of the molecular weight of the dCyms) it is possible that they are in fact measuring bivalent binding even at the lowest surface density.

To clearly demonstrate monomeric interactions, a simple order of addition experiment should be run. The authors could easily inject non-biotinylated ClpC1-NTD over the surfaces with homo-dCym captured on the biotin ClpC1-NTD surface. If the second site on the homo-dCym is not bound to a ClpC1-NTD on the surface, then they should see ClpC1-NTD binding. They should vary the amount of homo-dCym captured on the surface and since they know the mass of each reactant, they should be able to draw a linear correlation with the amount of ClpC1-NTD that can bind which would clearly demonstrate monomeric binding of the homodimeric dCym.

The above experiment brings up another issue the authors should address. In Figure 1b the authors depict ClpC1 binding to the homo-dCym and then sequentially binding to the complex. Yet in the text they state:

Using simplified desoxycyclomarin derivative 1 (Figure 1a) as a backbone, we aimed to develop homo-BacPROTAC molecules that simultaneously bind to two molecules of ClpC1.”

Which two molecules of ClpC1 are they referring to? Could homo-dCym simply dimerize the monomeric ClpC1 that is free in solution?

“We hypothesized that the crosslinking of ClpC1 units would induce “self-degradation” of this essential unfoldase (Figure 1b), leading to more efficient killing of mycobacteria as compared to the parental, non-degrading cyclomarins”

Could homo-dCym just crosslink two ClpC1s that are already in the ring complex?

Also, it might help to explain to non-experts in ClpC1 why the authors only studied binding to ClpC1-NTD? Is there an opportunity to study the ClpC1 complex? Are there any known structural differences between ClpC1-NTD in solution and within the complex which may afford an opportunity to create an actual chimeric molecule?

Given that the authors are deriving the KDs from the kinetics determined by SPR, the authors should report the kinetic values as well as the KDs. The kinetic information would be useful for anyone trying to replicate their work. It would be interesting to learn if the authors see any correlation with the KDs or kinetics with the “activity” of the molecule? Can they report if its better for example to have a compound with high affinity or a slow off rate. Or is it the case that the kinetics and/or affinity have no predictive correlation with potency in terms of antimicrobial activity?

The authors statement at line 194 is confusing “Exceptions to the latter observation were the two matched pairs 8/27 (entry 6) and 15/28 (entry 9), in which the unmethylated and methylated counterparts bind to ClpC1-NTD equally strong with KD values of 0.4 nM and 0.5 nM, respectively.” The KDs they report in Table 1 for 15/28 show values of 1.2 and 1.8 nM not 0.5 nM. Similarly, several of the KDs reported in Table 1 do not match the KDs reported for the same compounds in the Supplemental Table which is confusing.

Technical issues: Every numerical result in the manuscript should include an experimental standard deviation. The authors state that the SPR, cell free degradation and intracellular degradation assays each were run in triplicate and MIC assays were run in duplicate. Yet for all of these assays they only report an average value. The authors must include an experimental standard deviation with every reported value from any of the assays so that a reader can judge the variability in a particular assay.

And any numerical result should be reported to the correct number of significant digits. For example, in Table 1 three compounds share identical MIC values at 0.097 uM. Is this MIC assay capable of nM resolution? How is it possible that three different compounds yield the exact same result? Two other compounds were reported with MIC values of 3.13 and two with 6.25. It seems odd that so many of the MIC values are the same.

Similarly, the authors report at line 263 “BacPROTAC 8 (UdSBI-0545) induced degradation with an average DC50 of 571 nM and an average Dmax of 47.7% after 24 hours of incubation in a concentration-dependent manner.” Standard errors and correct number of units need to be included on all of these results. To a nonexpert in degradation assays, their reporting suggests this method also has a nM resolution in the DC50 along with 0.1% resolution in the Dmax.

Recommendation: The authors devote figures 2 and 3, and a hefty section in the start of the Results & Discussion to describing the synthesis of the compounds. Unless the chemical steps are unique, it would be better for the reader if these sections were moved to the supplemental.

REVIEWER COMMENTS

Reviewer #1 (Remarks to the Author):

Junk et al., in their manuscript entitled “BacPROTAC-induced degradation of ClpC1 as a new strategy against drug-resistant mycobacteria,” demonstrated for the first time the targeted protein degradation of an endogenous substrate in mycobacteria by the bacterial protease machinery, ClpC1P1P2.

This work is impactful because it shows homo dimeric binders of ClpC1, which is an essential component of the bacterial proteolytic machinery ClpC1P1P2, can also act as degraders to induce its own degradation. Previously, the original paper on the discovery of BacPROTACs by Morreale et al. (Cell 2022) provided the proof of concept that BacPROTACs can recruit the Clp-mediated degradation of non-endogenous model proteins of interest and fusion proteins in mycobacteria. In this manuscript, Junk et al. build on this impactful scientific premise to show how cyclomarins-based cyclic heptapeptides, possessing anti-Mtb-activity, with two ligands of ClpC1 direct the Clp protease machinery against its own ClpC1 subunit resulting in self-destruction.

This work has been thoroughly conducted, supporting its claims that homodimeric BacPROTACs exhibit improved anti-Mtb activity. Experiments supporting the claims include drug-resistant strains and in macrophages through experiments through degradation assays of ClpC1NTD comparing with ClpC1P2 and full-length ClpC1 both in cell-free assay systems and intracellular systems. Assessing MICs and MBCs against Msm #607, Mtb H37Rv strains to showcase these BacPROTACs not inhibiting gram +ve, gram -ve bacteria, or other NTMs.

We thank the reviewer for their positive comments.

Due to the noteworthy scientific findings, the manuscript is recommended for publication, despite some limitations in the physicochemical and PK properties of the BacPROTACs associated with solubility and bioavailability. The suggested minor revisions listed below would make the claims more robust or enable the reproducibility of the experimental data provided in this paper.

1. The following experimental data would bolster the claims against the activity of the BacPROTACs against drug-resistant strains of Mtb.

- Performing the assays against RIF-resistant strains (H37Rv rpoBS450L) since RIF is one of four drugs used in the first-line treatment of TB
- Standard isoniazid-resistant (INH: katGdel) strain (MS015), and a moxifloxacin-resistant (MOX: gyrAD94K) strains

To further strengthen the point that Homo-BacPROTACs kill drug resistant *Mtb*, we tested the degraders and the matching Cym monomers on the requested isoniazid resistant H37Rv (*katG*^{del}) ATCC 35822 (Rouse and Morris, *Infection and Immunity* 1995, 63, 1427-1433), the Rifampicin resistant H37Rv (RpoB^{S450L}) ATCC 35838 (Garcia et al., *J. Clin. Microbiol.* 2001, 39, 1813-1818), and a moxifloxacin resistant H37Rv clinical isolate, called FNDR-M1. Our data show that also for the tested mono-drug-resistant *Mtb* strains, Homo-BacPROTACs outperform their matching monomers, being more efficient in killing pathogenic mycobacteria as reflected by the lower MIC values. The new data are summarized in Table 2 and in the respective Materials and Methods part, where the culturing of all *Mtb* strains is described.

- Hypervirulent HN878 Mtb strain of the W/Beijing Lineage (Lineage 2), which is increasing in incidence in active TB cases and is frequently associated with the occurrence of drug resistance

We also tested the hypervirulent Beijing HN878 strain, which turned out to be highly sensitive to Homo-BacPROTACs, which outperformed their matching monomers in antibiotic activity. These data are summarized in Supplementary Figure 1.

- Listing the MIC values of the TB drugs, Streptomycin, RIF, INH, etc., in Table 3 instead of stating sensitive and resistant.

Thank you for pointing this out. The MIC values for these drugs have been added to the table.

2. Decimal points and commas are interchangeably used as decimal markers. One type of decimal marker needs to be used as a marker.

The decimal markers have been consistently changed to decimal points.

3. In the SI, listing the diagnostic peaks in ¹H and ¹³C NMR will be helpful for the readers in distinguishing major versus minor rotamer.

We agree with the reviewer on this point. For all compounds in which the rotamer signal sets could not be clearly distinguished, we added a section of selected diagnostic peaks.

4. In the SI ¹³C NMR, second decimal digits need to be listed to distinguish identical peaks; e.g., page 21 compound 7 has two peaks at 171.6 ppm.

We corrected all ¹³C NMR spectra to include second decimal digits for all cases where appropriate.

5. The literature reference needs to be cited for the known compounds in the intermediates for the enantiomeric BacPROTACs. If they are new compounds, full characterization data need to be provided.

We added the literature references for all known compounds and full characterization data for all intermediates that have not been previously described.

6. Many spectra contain impurities and solvent peaks for ethyl acetate, acetone, and dichloromethane. Cleaner spectra need to be provided for the compounds SI-7, 8, 9, 10, 15, 19, 21, 25, 26, 28, 30, 31, 36, 37 (poor resolution for ¹³C NMR), 47.

For all compounds mentioned, we now provide clean NMR spectra. For SI-37 we provide a ¹³C NMR spectrum with a higher signal to noise ratio.

Reviewer #2 (Remarks to the Author):

Cyclomarins (Cym), cyclic peptides produced by certain marine *Streptomyces* sp, are known to have antitubercular activity. They were shown to exert their toxicity by binding to the essential protease ClpC1. In their recent study, Junk et al redesign Cyms to not simply bind the target ClpC1 but instead cause its self-mediated degradation. The strategy is based on synthesis of a crosslinked Cym dimer. These compounds possess two ClpC1 binding entities such that ClpC1 protomers can be bound at one end and the ClpC1P degradation machine at the other, leading to proximity-induced degradation of the ClpC1 protomer by the ClpC1P complex.

The authors achieve this by introducing an alkyne moiety into the Cym molecule at specific positions, and then use a bifunctional crosslinker carrying two azide groups to ligate the two Cym molecules using azide-alkyne cycloaddition. The authors test different Cym derivatives with the aim to maximize eukaryotic cell permeability and biological activity. They can demonstrate that dimeric Cyms outperform their monomeric counterparts in their effect on Mtb, hence providing more potent compounds that address ClpC1 as a drug target.

The study is interesting and timely, and the protease self-targeting approach is novel. I support publication.

We thank the reviewer for their positive feedback.

However, several points should be addressed prior to publication:

1) PROTACS are usually bifunctional binders that feature one moiety that binds the target protein to be degraded and another moiety that binds the protease (or a ligase that tags the target protein for degradation). The Cym crosslinked dimers are not bifunctional, hence the term homo-BacPROTAC, since they serve the very specific task to target the protease for self-degradation. As such, Cym dimers do not represent a general strategy that could render any mycobacterial protein a degradation substrate, but they are ClpC1-specific drugs. Overstatements concerning the applicability should be avoided. The title should reflect the self-targeting nature of the approach. For example: "Homo-BacPROTAC-induced ClpC1-mediated self-degradation as a new strategy...".

We removed any statements that could be regarded as overstatements concerning the applicability.

We changed the title to "Homo-BacPROTAC induced degradation of ClpC1 as a strategy...". We now use the term "Homo-BacPROTAC" instead of "BacPROTAC" throughout the manuscript. We deliberately avoided a prominent use of the term "self-degradation", because ClpC1 is an unfoldase, not a protease, and therefore formally cannot degrade itself.

2) In vitro degradation assays: The reaction mixture was incubated for 90 min for each Cym concentration. Why was this time frame chosen? An actual time course should be shown for BacPROTAC 8 at the DC50 and at the highest tested concentration. Does it take 90 minutes to achieve degradation of half the ClpC1 N-terminal domain present, and when further time points are taken at 120 min and later, does the ClpC1 N-terminal domain eventually disappear completely or not?

In the course of assay development, we performed time course experiments, one of which is shown in Supplementary Figure 3. Although incubation times longer than 90 min lead to more complete removal of ClpC1-NTD, we picked 90 min as fixed time point to assess structure-activity relationships between different Homo-BacPROTACs and to monitor substantial potency improvements in the course of chemical diversification. The sensitivity of capillary Western allows detection of trace amounts (low pg range) for a given protein of interest. By this measure, ClpC1-NTD still can be detected even at the

highest concentrations of Homo-BacPROTACs and longest time points tested. There are several ways to explain this: The ATP regeneration system (required for ClpC1 activity) could be exhausted at extended times, dimeric degraders could bind in bivalent manner to ClpC1 oligomers once substrate is depleted, or the isolated ClpC1-NTD may become inaccessible with time due to partial misfolding. Please note that this assay was mainly instrumental to establish structure-activity relationships at the initial stage of the project, for example when searching for best linker attachment points etc.

3) The effect of binding-competent Cym monomers (5 and dCymC) on ClpC1 in vivo levels should also be tested and included in Figure 5 to show that the degradation observed in vivo is mediated only by the dimeric Cyms and not the monomeric Cyms.

We performed the suggested experiments. Figure 5 and the corresponding text have been revised to include data for the monomers **5**, **8** and **dCymC**. No degradation of ClpC1 was observed for these compounds.

4) The core result of the study is the strongly improved MIC compared to smaller changes in the binding of ClpC1 for the dimeric versus monomeric Cyms, highlighting the power of harnessing an enzyme instead of relying on binding. It is therefore concerning that this appears to only hold true for *Mtb* and not for any of the other mycobacteria. For *M. smegmatis* the MICs for the dimeric and monomeric Cyms were comparable and for other pathogenic mycobacteria, only the monomeric Cyms showed growth inhibition.

Given the high homology between mycobacterial ClpC1, how can this be explained? Have the authors assessed the conservation of the known Cym binding residues in ClpC1 orthologs? Furthermore, if monomeric Cyms bind, why would the dimeric Cym not work? The authors should test the permeability of monomeric and dimeric Cyms into the different mycobacterial organisms to assess if their cell walls present different barriers.

The reviewer raises an important point. The cyclomarin receptor domains, the ClpC1 NTDs, are 100% conserved in the mycobacterial strains (*Mtb*, *Msm* and NTM) assessed. As correctly pointed out, some nontuberculous mycobacteria do not, or only weakly, respond to the Homo-BacPROTAC ClpC1 degraders. A likely explanation is provided in our accompanying study (Hoi *et al.*, *Cell* 2023, 186, 2176-2192): In response to Cym antibiotics, which present a small-molecule mimic of misfolded proteins, mycobacteria upregulate a Clp specific security system, preventing an overload of the ClpC1P1P2 protease. This protection system is mainly composed of the ClpC2 protein featuring a functional Cym (and thus also Homo-BacPROTAC) binding site. Mycobacteria lacking this system are more sensitive towards antibiotics targeting ClpC1. Upregulation of ClpC2 varies drastically between different mycobacteria, providing *Msm* a much better protection than *Mtb*, which could explain enhanced sensitivity of *Mtb* towards Homo-BacPROTACs.

In addition to mounting different ClpC2 responses, mycobacteria react differently to environmental and nutritional stresses. Uncovering the sensing and integrating mechanism that ultimately coordinate protein quality control is an important direction of future research. For example, it needs to be resolved how quality control machineries maintain protein homeostasis in dormant states and which specific adaptations and feedback mechanisms have been accomplished in the different mycobacteria.

With regards to the bacterial drug uptake, standard operations did not work with Cym-like compounds, presumably due to their extreme hydrophobicity and their absorption by the cell wall. We tried to establish LC-MS based detection of dCymC, Homo-BacPROTACs and monomers after incubation in

Msm cells, but could not optimize the method to allow reproducible, sensitive detection of compounds in the cytosol, despite extensively testing lysis and extraction methods. We therefore cannot answer the direct drug uptake question of Homo-BacPROTACs versus monomers in mycobacteria. However, we believe that the functional readouts, i.e. the cognate target degradation of ClpC1 in *Msm* and *Mtb* cells proves that the Homo-BacPROTACs enter the respective cells.

Minor comment:

The labels in Figure 4 are too small.

We increased the size of the labels accordingly.

Reviewer #3 (Remarks to the Author):

This manuscript is very interesting to show Homo-BacPROTACs for degrading ClpC1. I have some concerns:

1. The binding affinity is very high for the dimer in low nanomolar range even picomolar, however, the degradation is much lower reflecting by the DC₅₀ and D_{max} in high micromolar. Whereas, inhibition of the *Mtb* growth is much better for 12 (97 nM). My concern is: whether the degradation is predominated for the biological activity? Or other mechanism in the anti-bacteria activity?

The reviewer raises a valid concern. We could demonstrate in *Msm* by capillary Western that ClpC1 is being degraded in the single digit μ M range with the Homo-BacPROTACs **8** and **12**. Their DC₅₀ / activity fits well to the single-digit to low double-digit μ M MIC elicited by the compounds in this strain. In *Mtb* H37Rv, we could demonstrate in our accompanying paper that at a single concentration (7.8 μ M for 48 h) strong degradation of *Mtb* ClpC1 could be achieved (see Figure 7B and 7C of Hoi *et al.*, *Cell* 2023, 186, 2176-2192). Since the counteracting activity of the ClpC2/ClpC3 feedback loop is much less predominant in *Mtb* than in *Msm*, we have proposed that the lower amounts of ClpC2 in *Mtb* is insufficient to buffer the degrader, resulting in efficient killing of *Mtb* H37Rv as reflected by the lower MIC (please see also discussion at Reviewer2/Point4). The proteomics analyses in Hoi *et al.*, *Cell* 2023, 186, 2176-2192 also demonstrate that Homo-BacPROTAC treatment induces drastic changes in the overall proteomic spectrum, leading to the up- and downregulation of multiple mycobacterial proteins. We interpret this observation that Homo-BacPROTACs, in addition to inducing ClpC1 degradation, inherit the cytotoxicity of the incorporated cyclomarin derivative, which interfere with ClpC1-mediated protein degradation. Thus, the Homo-BacPROTACs benefit from a dual activity, simultaneously deregulating and destroying an essential component of the mycobacterial protein quality control system.

2. As the author mentioned, homo-PROTAC has several limitations, e.g. physicochemical properties, solubility, polarity, bioavailability Therefore, what are the strengths for the homo-PROTAC? Only a new concept? How about the in vivo efficacy? What is the difference for homo-PROTAC compared with biological method?

We did not evaluate the in vivo efficacy of Homo-BacPROTACs in an animal model because of their poor pharmacokinetic properties in the eukaryotic system. We would like to point out that the molecules shown represent a promising starting point, but require further chemical optimization to convert them into drugs useful for patients in need.

Substantial effort in chemical modifications or formulation would be needed to achieve this, however these highly ambitious tasks go beyond the scope of our current study, providing the proof-of-concept that Homo-BacPROTACs can target mycobacteria within eukaryotic cells. We presume that the concept of reprogramming ClpC1 will serve as a blueprint for the development of bacterial TPD agents, targeting different types of Clp proteases, thus extending the approach to the entire bacterial kingdom.

We are not quite sure, what the expression “compared with biological method” refers to. If it refers to the natural mechanism of how substrates are being removed by the ClpC1P1P2 degradation machinery, we kindly refer to Hoi *et al.*, *Cell* 2023, 186, 2176-2192 with respect to cyclomarin-based mechanisms. For the pArg/McsB-mediated degradation mode of action, we kindly refer to Trentini *et al.*, *Nature* 2016, 539, 48-53 and Ogbonna *et al.*, *Microbiology Spectrum* 2022, e02042-22.

Reviewer #4 (Remarks to the Author):

This important manuscript examines Bac-PROTACs as a new strategy for treating drug-resistant tuberculosis. Targeted protein degradation is an important emerging technology, which has thus far primarily been applied to Eukaryotic systems by manipulating the ubiquitin-proteasome degradation system to destroy rather than merely inhibit therapeutic targets. In this manuscript, ClpC1 targeting BacPROTACs are generated and examined starting from cyclomarin, a natural product that binds ClpC1 NTD. The most potent compounds produced (homodimeric-cyclomarin analogs) show potent anti-tuberculosis activity including against multi-drug resistant strains, an exciting result.

This chemistry-focused manuscript is highly rigorous, with extensive biological and chemical data presented in the main manuscript, in the supplementary data section, and in an associated manuscript under review, giving high confidence in the reliability and reproducibility of the results presented. The chemistry design for the series generation is well rationalized and easy to follow, with the consideration of structural changes to increase the permeability and pharmacological properties of the series particularly well conveyed.

We thank the reviewer for their supportive comments.

The primary weakness of this manuscript is that there is not adequate validation that the ClpC1 NTD is the target of the Bac-PROTACs 8 and 12, within the cell. Proteomic experiments in the associated manuscript suggest that this is true, but the routine methods for demonstrating on-target inhibition in *Mtb* through resistant mutant generation and sequencing, or ClpC1 binding site mutagenesis, have not been explored. This is a concern, as the introduction highlights the importance of antimicrobial resistance and the need for new therapeutic approaches, yet the propensity for resistance development for this new therapeutic class is not adequately considered.

The referee raises important points. With regards to Homo-BacPROTAC resistance mutations, this point was addressed during the revision of our accompanying paper (Hoi *et al.*, *Cell* 2023, 186, 2176-2192). In brief, we identified two resistance mutations (under laboratory in vitro conditions), both of which were located in the Cym binding site of ClpC1. These experiments clearly show that ClpC1 is the key cellular target of Homo-BacPROTACs in *Mtb* (Hoi *et al.*, *Cell* 2023, 186, 2176-2192, Figure 7D,E). Regarding known resistance mutations against other *Mtb* drugs, we tested the efficacy of Homo-BacPROTACs against rifampicin-, moxifloxacin- and isoniazid-resistant, as well as hypervirulent *Mtb* strains (Table 2, Supplementary Figure 1). In all cases, the Homo-BacPROTACs outperformed the corresponding monomers and retained their activity against these strains, highlighting their pronounced antibacterial activity (please see also comments to Reviewer1/Point1).

In summary, some additional MOA evidence is needed in this otherwise exceptional manuscript, which is judged as highly suitable for Nature Communications.

Further suggestions:

1. The anti-tubercular activity of 8 and 12 should also be examined against non-growing *Mtb* cells as this is a major therapeutic need.

To induce non-growing conditions of *Mtb* cells, several approaches are possible. We used starvation for 3 weeks to induce a non-growing state of *Mtb* H37Rv. Under the conditions tested, the Homo-BacPROTACs did not show an inhibitory effect, while the monomers could reduce growth at least to

some extent at high concentrations. We describe these findings now in the main text, added Supplementary Figure 2 and adapted the respective methods section in the Supplementary Information. To further put this finding into perspective: starved cells become highly tolerant to almost everything, so this is a very stringent model. While bedaquiline retains certain partial growth inhibition properties against starved cells, its effect against hypoxia-induced persistent cells of *Mtb* was reported to be much more potent. So the hypoxia-induced model of persistence would likely be less stringent, but is technically not feasible given the time constraints. In addition, we also refer to other conditions we recently published in Hoi *et al.*, *Cell* 2023, 186, 2176-2192 where dormancy was induced by ATP depletion. Under those conditions, the Homo-BacPROTACs stay active.

The fact that Homo-BacPROTACs have a reduced activity in some mycobacteria (see NTM strains, Supplementary Table 10), or under certain experimental conditions could have several explanations.

As discussed previously (Reviewer2/Point4), the ClpC2 system that prevents an overload of the ClpC1P1P2 protease and secures the system against antibiotics (Hoi *et al.*, *Cell* 2023, 186, 2176-2192), seems to be differently regulated in the different mycobacteria, providing either less (*Mtb*) or more (*Msm*) protection against Homo-BacPROTAC degraders. Moreover, the ClpC2 response is co-regulated with the heat-shock response and other proteotoxic stresses, making it impossible to predict the efficacy of the respective rescue mechanism and thus potency of antibiotics. To understand the role of ClpC1 safeguard proteins in antibiotic and proteotoxic stress response pathways, a whole series of further experiments in different mycobacterial strains is required, going beyond the scope of this publication.

We have added statements to the conclusion (line 380 - line 389) which point to these or similar limitations of the Homo-BacPROTACs.

2. Line 112 – “we show” or if we showed provide the citation

We thank the reviewer for this comment, we changed the wording to “we show” (now line 114)

3. Line 195 - Why Caco-2 and not PAMPA when considering cellular permeability, these molecules are highly unlikely to be oral agents?

In general, orally bioavailable drugs are highly desirable for TB treatment. Despite the large size of the Homo-BacPROTACs, we aimed to assess their permeability in the less artificial Caco-2 permeability assay to gain insights into structure-activity relationships and find starting points for further modifications of the cyclomarin backbone.

4. Line 214 – The statement is incorrect the vectors have lower AUCs and higher clearance than the corresponding BacPROTACs

We have to apologize for the misunderstanding: We have used the word *vector* only to specify the linker attachment position (exit vector). Compounds **6** and **8** use the same exit vector (position 6) and show higher AUC and longer half times than BacPROTAC **12** (exit vector in position 7) or monomer **5**. We clarified this by providing compound numbers in the corresponding part of the text (now line 217).

5. Line 252 – Please rationalize why the BacPROTACs degradation is at a much higher concentration than the *Mtb* MIC – does this suggest there is another target?

The reviewer raises an important question that we have addressed in previous comments. Please see comments at Reviewer2/Point4, and Reviewer3/Point1.

6. Figure 6 graphs are very hard to read

The graphs have been changed to enhance readability.

7. SPR sensorgrams should be provided in full in the supplementary data section for all key molecules
SPR sensorgrams have been added for each key molecule (Supplementary Figure 7).

8. Table 3 MIC values for the control antibiotics (Strep, INH< Rif, Ami,) determined at the same time as homo-bacPROTACs should be reported as numerical values

We now show the MIC values for all control antibiotics in the table.

9. Control compound values are missing from the supplementary data table S1 for instance high/medium/low Caco-2 controls – a moderate concern

The Caco-2 assay we used is a well-established assay at Boehringer Ingelheim and is regularly validated with control compounds.

10. Raw data for the IV PK testing should be provided in either a data table or graphic form

We now provide the raw data of the IV PK experiments in Supplementary Tables 4-7.

11. S15 General information (chemistry), please provide a statement on how compound purity was determined

We added the following statement to the general information section: *The purity of final compounds was determined via analytical HPLC (column: Luna 3 μ m C18(2), 50x4.6 mm; flow: 1 mL min⁻¹; MeCN/H₂O gradient).*

Reviewer #5 (Remarks to the Author):

Key results: The authors present the synthesis of homodimeric forms of cyclomarins and demonstrate that these compounds may lead to the degradation of ClpC1 and antibacterial activity. Significance to the field: As a nonexpert in the field of antimicrobials it is difficult to judge the significance and novelty of the molecules themselves which were previously known to bind ClpC1 as monomers.

The editors requested input on the application of SPR. It is curious to see the authors present 200 pages of supplemental data which include 196 figures of NMR data sets but not a single SPR data set. The supplemental should include figures of all the SPR data overlaid with fits to the model used to extract the binding constants.

We have added SPR sensorgrams for each key molecule to the Supplementary Information (Supplementary Figure 7), including the fits we used to extract binding constants.

In order for others to be able to reproduce the SPR assays, the amount of target captured for each study should also be included with the figure along with a list of the analyte concentrations that were used in each injection.

The amount of target captured is indicated in each sensorgram (Supplementary Figure 7). A list of the analyte concentrations injected can be found in the figure legend.

The authors should also clearly state what they used for the molar concentrations of the dimeric compounds compared to the monomeric forms.

We used the same molar concentrations for both monomeric compounds and Homo-BacPROTACs for the evaluation of all SPR experiments. We have now tried to describe our method more precisely.

The authors claim the increase in the binding affinity for the homodimeric forms of dCym compared to monomeric forms is not due to avidity. However, the method the authors report using (comparing KD vs surface density) is not conclusive. Since they are using a CM5 surface (which is coated with a flexible dextran layer) and they are limited in how low they can go in surface density (because of the molecular weight of the dCyms) it is possible that they are in fact measuring bivalent binding even at the lowest surface density.

To clearly demonstrate monomeric interactions, a simple order of addition experiment should be run. The authors could easily inject non-biotinylated ClpC1-NTD over the surfaces with homo-dCym captured on the biotin ClpC1-NTD surface. If the second site on the homo-dCym is not bound to a ClpC1-NTD on the surface, then they should see ClpC1-NTD binding. They should vary the amount of homo-dCym captured on the surface and since they know the mass of each reactant, they should be able to draw a linear correlation with the amount of ClpC1-NTD that can bind which would clearly demonstrate monomeric binding of the homodimeric dCym.

The experiment suggested should be able to address the question raised by the reviewer. We actually performed the described experiment in the past. However, results were not conclusive. We realized that interpretation of the data was complicated by the fact that the injected protein could remove surface-bound Homo-BacPROTAC from the chip. This effect was observed over the whole range of concentrations we tested.

The above experiment brings up another issue the authors should address. In Figure 1b the authors depict ClpC1 binding to the homo-dCym and then sequentially binding to the complex. Yet in the text they state:

Using simplified desoxycyclomarin derivative 1 (Figure 1a) as a backbone, we aimed to develop homo-BacPROTAC molecules that simultaneously bind to two molecules of ClpC1.”

Which two molecules of ClpC1 are they referring to? Could homo-dCym simply dimerize the monomeric ClpC1 that is free in solution?

“We hypothesized that the crosslinking of ClpC1 units would induce “self-degradation” of this essential unfoldase (Figure 1b), leading to more efficient killing of mycobacteria as compared to the parental, non-degrading cyclomarins”

Could homo-dCym just crosslink two ClpC1s that are already in the ring complex?

The sentences in question explain the proposed mode of action of the homodimeric compounds. We would like to see binding of a ClpC1 monomer on one end of the BacPROTAC and binding to a ClpC1 as part of a hexameric complex on the other end to facilitate degradation of the former. However, binding between two monomeric ClpC1 molecules or even two ClpC1 molecules within a hexameric complex or as parts of two individual hexameric complexes cannot be excluded. In terms of experiments, this effect is relevant for *in vitro* (and *in vivo*) experiments but not for SPR, because ClpC1 NTD cannot oligomerize.

Also, it might help to explain to non-experts in ClpC1 why the authors only studied binding to ClpC1-NTD? Is there an opportunity to study the ClpC1 complex? Are there any known structural differences between ClpC1-NTD in solution and within the complex which may afford an opportunity to create an actual chimeric molecule?

While we did perform SPR experiments using full-length ClpC1, we encountered difficulties regarding data evaluation. Firstly, the oligomeric state of ClpC1 on the chip was difficult to control. Recombinant full-length ClpC1 exists in an equilibrium of different oligomeric states. We used up-front gel filtration experiments to find conditions, under which ClpC1 full-length existed predominantly in a hexameric complex. We used these conditions for immobilization of the protein and assumed that it existed predominantly in a hexameric state on the chip. Secondly, kinetics become complex and unfortunately, it was not possible to fit the data in a meaningful way.

With regards to structural data, it should be noted that the full-length ClpC1 is a rather dynamic complex, switching between monomer, hexamer and multimers of the hexamer. The latter higher-order complexes, which present the active state of the unfoldase, are formed upon binding of Cym derivatives or Homo-BacPROTACs, or when sequestering misfolded proteins. The NTD that binds Cym and Homo-BacPROTACs exhibits an inherent en-bloc mobility in the activated complex, such that it cannot be visualized by cryo-EM or crystallographic methods. As pointed out by the reviewer, having access to a well-defined NTD in an active complex would be extremely valuable to characterize the exact binding mode of Homo-BacPROTACs, estimate ClpC1 contacts from protein portions outside the NTD and analyze the bivalent binding of Homo-BacPROTACs by adjacent NTDs in the oligomer. The latter knowledge, for example, would be helpful to design Homo-BacPROTAC derivatives that cannot undergo intra-hexamer contacts and bind preferentially to ClpC1 subunits in trans (i.e. degrader substrates).

Given that the authors are deriving the K_D s from the kinetics determined by SPR, the authors should report the kinetic values as well as the K_D s. The kinetic information would be useful for anyone trying to replicate their work. It would be interesting to learn if the authors see any correlation with the K_D s or kinetics with the “activity” of the molecule? Can they report if its better for example to have a compound with high affinity or a slow off rate. Or is it the case that the kinetics and/or affinity have no predictive correlation with potency in terms of antimicrobial activity?

We can see a rough correlation between K_D and the MIC activity against *Mtb* H37Rv (Supplementary Tables 1 and 2). We do also see a correlation between K_D and off-rate. It is commonly observed in drug molecules that the affinity is off-rate driven. Since there is no additional knowledge gain obtained from these data points, we will only include the off-rates in Supplementary Table 1.

The authors statement at line 194 is confusing “Exceptions to the latter observation were the two matched pairs 8/27 (entry 6) and 15/28 (entry 9), in which the unmethylated and methylated counterparts bind to ClpC1-NTD equally strong with K_D values of 0.4 nM and 0.5 nM, respectively.” The K_D s they report in Table 1 for 15/28 show values of 1.2 and 1.8 nM not 0.5 nM. Similarly, several of the K_D s reported in Table 1 do not match the K_D s reported for the same compounds in the Supplemental Table which is confusing.

We thank the referee for pointing this out. We have now corrected the discrepancies between the two tables and changed the sentence in question (line 198).

Technical issues: Every numerical result in the manuscript should include an experimental standard deviation. The authors state that the SPR, cell free degradation and intracellular degradation assays each were run in triplicate and MIC assays were run in duplicate. Yet for all of these assays they only report an average value. The authors must include an experimental standard deviation with every reported value from any of the assays so that a reader can judge the variability in a particular assay.

We added standard deviations to all reported values if applicable.

And any numerical result should be reported to the correct number of significant digits. For example, in Table 1 three compounds share identical MIC values at 0.097 μ M. Is this MIC assay capable of nM resolution? How is it possible that three different compounds yield the exact same result? Two other compounds were reported with MIC values of 3.13 and two with 6.25. It seems odd that so many of the MIC values are the same.

We now report every numerical result with the correct number of significant digits.

The minimum inhibitory concentration is defined as the minimal concentration needed to completely inhibit growth. In our experimental setup, each well contains a distinct compound concentration, either done in duplicates or triplicates. Briefly, the test compounds were dissolved in DMSO, serially diluted by 2-fold in a 10-concentration dose response ranging from 128 to 0.25 μ g/mL in 96-well plates. The concentration in the first well displaying no more growth (i. e. the minimum concentration) is reported as the MIC value. If two compounds show 100% inhibition in wells containing the same compound concentration, the MIC value is identical.

Similarly, the authors report at line 263 “BacPROTAC 8 (UdSBI-0545) induced degradation with an

average DC50 of 571 nM and an average Dmax of 47.7% after 24 hours of incubation in a concentration-dependent manner..” Standard errors and correct number of units need to be included on all of these results. To a nonexpert in degradation assays, their reporting suggests this method also has a nM resolution in the DC50 along with 0.1% resolution in the Dmax.

We thank the reviewer for pointing this out. We have changed all DC₅₀ and D_{max} values to two significant digits and have generally adjusted all reported values to the correct number of significant digits.

Recommendation: The authors devote figures 2 and 3, and a hefty section in the start of the Results & Discussion to describing the synthesis of the compounds. Unless the chemical steps are unique, it would be better for the reader if these sections were moved to the supplemental.

After consultation with the editor, we agreed to leave the sections about the chemical syntheses and the figures in the manuscript. We believe that these are of particular interest to readers with a chemistry background.

REVIEWER COMMENTS

Reviewer #1 (Remarks to the Author):

All the concerns have been adequately addressed. Recommended for publication as is.

Reviewer #2 (Remarks to the Author):

The authors have addressed the raised concerns. I support publication in Nature Communications.

Reviewer #3 (Remarks to the Author):

The author provides reasonable explanations on my concerns, however, i still suggest to include in vivo strengths for the homoPROTACs to give some image that the protac is just a new compound or a new chemical konckdown tool?

Reviewer #4 (Remarks to the Author):

The authors have adequately addressed all the concerns and suggestions of this reviewer, including those in relation to resistance development. The manuscript is impactful and will be a welcome addition the nascent field of targeted protein degradation in bacteria.

For the comments of Reviewer 5, please see the attachment.

Reviewer's original comments in black

Authors' responses in blue

Reviewer's follow up comment in red.

The authors claim the increase in the binding affinity for the homodimeric forms of dCym compared to monomeric forms is not due to avidity. However, the method the authors report using (comparing KD vs surface density) is not conclusive. Since they are using a CM5 surface (which is coated with a flexible dextran layer) and they are limited in how low they can go in surface density (because of the molecular weight of the dCyms) it is possible that they are in fact measuring bivalent binding even at the lowest surface density. To clearly demonstrate monomeric interactions, a simple order of addition experiment should be run. The authors could easily inject non-biotinylated ClpC1-NTD over the surfaces with homo-dCym captured on the biotin ClpC1-NTD surface. If the second site on the homo-dCym is not bound to a ClpC1-NTD on the surface, then they should see ClpC1-NTD binding. They should vary the amount of homo-dCym captured on the surface and since they know the mass of each reactant, they should be able to draw a linear correlation with the amount of ClpC1-NTD that can bind which would clearly demonstrate monomeric binding of the homodimeric dCym.

The experiment suggested should be able to address the question raised by the reviewer. We actually performed the described experiment in the past. However, results were not conclusive. We realized that interpretation of the data was complicated by the fact that the injected protein could remove surface-bound Homo-BacPROTAC from the chip. This effect was observed over the whole range of concentrations we tested.

Failure to see a binding response for ClpC1-NTD demonstrates the homodimeric forms of dCym are cross linking with ClpC1-NTD that is on the surface. A description of this type of experiment should be included in the manuscript. Given that the authors claim the KDs for the monomeric binding of a dCym compound are ~ 1 nM, an injection of ClpC1-NTD at the KD concentration of 1 nM over a surface with homodimeric dCym should bind up half the sites on the surface and give a very large binding response. The authors observations that "injected protein could remove surface-bound Homo-BacPROTAC from the chip" is related to mass transport effects and not to active displacement. In order for ClpC1-NTD to actively displace the Homo-BacPROTAC from the bound compound with ClpC1-NTD on the sensor surface it would in fact need to be binding to an allosteric site. ClpC1-NTD in solution is actually acting as a sink to block the rebinding of Homo-BacPROTAC to the surface. The authors should include in the test something along the lines of...*"To demonstrate that homodimeric forms of dCym were binding monovalently to the ClpC1-NTD surface, in the past we did the following stacking experiment. In the first injection step, we captured a monomer form of Cym and a dimer form of Cym onto separate ClpC1-NTD surfaces. In a second injection, we introduced non biotinylated ClpC1-NTD in solution. We observed no binding of ClpC1-NTD to either form of Cym captured on the ClpC1-NTD surfaces. The simplest interpretation of this result is that the homodimeric dCym is bound bivalently to the ClpC1-NTD on the sensor surface."*

Given that the authors are deriving the KDs from the kinetics determined by SPR, the authors should report the kinetic values as well as the KDs. The kinetic information would be useful for anyone trying to replicate their work. It would be interesting to learn if the authors see any correlation with the KDs or kinetics with the "activity" of the molecule? Can they report if its better for example to have a compound with high affinity or a slow off rate. Or is it the case that the kinetics and/or affinity have no predictive correlation with potency in terms of antimicrobial activity?

We can see a rough correlation between KD and the MIC activity against Mtb H37Rv (Supplementary Tables 1 and 2). **We do also see a correlation between KD and off-rate.** It is commonly observed in drug molecules that the affinity is off-rate driven. Since there is no additional knowledge gain obtained from these data points, we will only include the off-rates in Supplementary Table 1.

Of course, you will see a correlation between KD and off-rate since the KD is dependent on the off-rate. Since the authors reporting KD determined from a kinetic analysis of the data, the on-rate must be included in the results Table along with the off-rate. The authors' suggestion that no additional knowledge is gained from listing the on-rates is not correct. They may not be able to the importance of including this information for the reader.

Technical issues: Every numerical result in the manuscript should include an experimental standard deviation. The authors state that the SPR, cell free degradation and intracellular degradation assays each were run in triplicate and MIC assays were run in duplicate. Yet for all of these assays they only report an average value. The authors must include an experimental standard deviation with every reported value from any of the assays so that a reader can judge the variability in a particular assay.

We added standard deviations to all reported values if applicable.

In the original manuscript, the authors stated that SPR assays were run in triplicate. However, in this updated data draft they now state that "At least two independent measurements were performed for each compound". Why were some data sets left out in updated draft? The authors should add information to the Table which indicates the number of assays for a particular compound.

The authors should also clearly state what they used for the molar concentrations of the dimeric compounds compared to the monomeric forms.

We used the same molar concentrations for both monomeric compounds and Homo-BacPROTACs for the evaluation of all SPR experiments. We have now tried to describe our method more precisely.

Since the authors are using molar concentration of compound, they should explain how doubling the concentration of the Cym binding sites for dimeric compounds will impact the on rate compared to the monomeric forms. All things being equal, would a 100 nM solution of monomer display the same apparent binding rate as a 100 nM solution of dimer or a 50 nM solution of dimer?

And any numerical result should be reported to the correct number of significant digits. For example, in Table 1 three compounds share identical MIC values at 0.097 μ M. Is this MIC assay capable of nM resolution? How is it possible that three different compounds yield the exact same result? Two other compounds were reported with MIC values of 3.13 and two with 6.25. It seems odd that so many of the MIC values are the same.

We now report every numerical result with the correct number of significant digits.

Supplementary Tables 1-11

Supplementary Table 1. Structure-Activity Relationships part 1 (SPR, Caco-2)

compound ¹	Compound descriptor	K _o ClpC1-NTD ² [nM] Mean ± SD	K _o correlation ³	k _{off} ClpC1-NTD ⁴ [s ⁻¹] Mean ± SD	Caco-2 permeability ⁵				Permeability Correlation ⁵	
					P (A-B)	P (B-A)	Efflux ratio	Int. Perm	P (A-B)	Ratio
SI-38	[3-OPra]	5.6 ± 0.6	2	5.3E-02 ± 1.5E-02	<0.2	18		9.1	>75	>31
SI-64	[6-Me][3-OPra]	11.3 ± 1.1		1.6E-01 ± 1.5E-01	15	43	2.9	29.0		

It is always good to get in the habit of reporting experimental standard deviations and reporting them in a way that is meaningful to the reader. The number of significant digits is determined by the experimental standard deviation. For example, with the SPR data 5.6±0.6 is correct. But 11.3±1.1 should be reported as 11±1.

5.3E-02±1.5E-02 should be reported as 5E-02±2E-02

1.6E-01±1.5E-01 should be reported as 2E-01±2E-01 What does it mean to have an off rate with a standard error as big as the off rate itself? Does that tell us anything about results?

A minor point is the authors would save some clutter in the Table by not including the -0. None of their off rates need two units in the exp.

So, they could just write 5E-02±2E-02 as 5E-2±2E-2. Or better yet (5±2)E-2

The authors should include an explanation as to why different amounts of dissociation phase data appear to be analyzed for different compounds. Many data sets seem to be fit out to 1000 seconds but others (typically the ones with a slower dissociation phase) seem to be missing dissociation phase data as highlighted below. Why were these data deleted?

8

9

10

11

12

13

14

15

16

17

20

21

106

99

105

Reviewer's original comments in black

Authors' responses in blue and purple for second response letter

Reviewer's follow up comment in red

REVIEWER #3:

"...The author provides reasonable explanations on my concerns, however, i still suggest to include in vivo strengths for the homoPROTACs to give some image that the protac is just a new compound or a new chemical konckdown tool?..."

"...I mean I still would love to see the in vivo efficacy of homoPROTAC. If the TAC is only a chemical tool, I do not think it is a useful compound for further development..."

We did not evaluate the in vivo efficacy of Homo-BacPROTACs in an animal model because of their poor pharmacokinetic properties and essentially non-existent oral bioavailability in the rodent organism. Even if repeated i.v administration might have been a path forward, we would not conduct such animal experiments in line of the fact that a Homo-BacPROTAC drug would have to be applied orally in MDR tuberculosis patients. An i.v -based animal model therefore has only very limited predictive value. We would like to point out that the molecules shown represent a promising starting point highlighting a novel antibiotic concept.

As stated in the previous rebuttal, substantial efforts in chemical modifications or formulation would be needed to improve pharmacokinetic properties, however these highly ambitious tasks go beyond the scope of our current study. Thus, we agree that as of today the Homo-BacPROTACs can formally be considered as chemical tools, but we also emphasize that the reported compounds could be further optimized and that the concept per se is not a dead end. Accordingly, we have added this statement to the Conclusion in the main text:

"...However, the reported Homo-BacPROTACs have several limitations, primarily with respect to their physicochemical properties, such as solubility and total polar surface area. This limits their bioavailability and pharmacodynamic profiling in vivo and will require further optimization. Our preliminary SAR studies describe potential strategies going forward and opportunities for synthesis of the next generation of Homo-BacPROTACs..."

REVIEWER #5:

The authors claim the increase in the binding affinity for the homodimeric forms of dCym compared to monomeric forms is not due to avidity. However, the method the authors report using (comparing KD vs surface density) is not conclusive. Since they are using a CM5 surface (which is coated with a flexible dextran layer) and they are limited in how low they can go in surface density (because of the molecular weight of the dCyms) it is possible that they are in fact measuring bivalent binding even at the lowest surface density. To clearly demonstrate monomeric interactions, a simple order of addition experiment should be run. The authors could easily inject non-biotinylated ClpC1-NTD over the surfaces with homo-dCym captured on the biotin ClpC1-NTD surface. If the second site on the homo-dCym is not bound to a ClpC1-NTD on the surface, then they should see ClpC1-NTD binding. They should vary the amount of homo-dCym captured on the surface and since they know the mass of each reactant, they should be able to draw a linear correlation with the amount of ClpC1-NTD that can bind which would clearly demonstrate monomeric binding of the homodimeric dCym.

The experiment suggested should be able to address the question raised by the reviewer. We actually performed the described experiment in the past. However, results were not conclusive. We realized that interpretation of the data was complicated by the fact that the injected protein could remove surface-bound Homo-BacPROTAC from the chip. This effect was observed over the whole range of concentrations we tested.

Failure to see a binding response for ClpC1-NTD demonstrates the homodimeric forms of dCym are cross linking with ClpC1-NTD that is on the surface. A description of this type of experiment should be included in the manuscript. Given that the authors claim the KDs for the monomeric binding of a dCym compound are ~ 1 nM, an injection of ClpC1-NTD at the KD concentration of 1 nM over a surface with homodimeric dCym should bind up half the sites on the surface and give a very large binding response. The authors observations that “injected protein could remove surface-bound Homo-BacPROTAC from the chip” is related to mass transport effects and not to active displacement. In order for ClpC1-NTD to actively displace the Homo-BacPROTAC from the bound compound with ClpC1-NTD on the sensor surface it would in fact need to be binding to an allosteric site. ClpC1-NTD in solution is actually acting as a sink to block the rebinding of Homo-BacPROTAC to the surface. The authors should include in the test something along the lines of...*“To demonstrate that homodimeric forms of dCym were binding monovalently to the ClpC1-NTD surface, in the past we did the following stacking experiment. In the first injection step, we captured a monomer form of Cym and a dimer form of Cym onto separate ClpC1-NTD surfaces. In a second injection, we introduced non biotinylated ClpC1-NTD in solution. We observed no binding of ClpC1-NTD to either form of Cym captured on the ClpC1-NTD surfaces. The simplest interpretation of this result is that the homodimeric dCym is bound bivalently to the ClpC1-NTD on the sensor surface.”*

We agree that the most likely explanation for the phenomenon observed was crosslinking of two NTD molecules by bifunctional BacPROTAC molecules on the chip. We have included a description of our experiment in the manuscript. We were always aware of the fact that the situation on the SPR chip is a very artificial one. In our biophysical experiments we are using a small portion of the full-length protein, which forms different oligomeric states in vivo, offering a plethora of potential intermolecular interactions. This, of course, needs to be kept in mind when comparing in vitro results with in vivo data.

Given that the authors are deriving the KDs from the kinetics determined by SPR, the authors should report the kinetic values as well as the KDs. The kinetic information would be useful for anyone trying to replicate their work. It would be interesting to learn if the authors see any correlation with the KDs or kinetics with the “activity” of the molecule? Can they report if its better for example to have a compound with high affinity or a slow off rate. Or is it the case that the kinetics and/or affinity have no predictive correlation with potency in terms of antimicrobial activity?

We can see a rough correlation between KD and the MIC activity against Mtb H37Rv (Supplementary Tables 1 and 2). We do also see a correlation between KD and off-rate. It is commonly observed in drug molecules that the affinity is off-rate driven. Since there is no additional knowledge gain obtained from these data points, we will only include the off-rates in Supplementary Table 1.

Of course, you will see a correlation between KD and off-rate since the KD is dependent on the offrate. Since the authors reporting KD determined from a kinetic analysis of the data, the on-rate must be included in the results Table along with the off-rate. The authors’ suggestion that no additional knowledge is gained from listing the on-rates is not correct. They may not be able to the importance of including this information for the reader.

We thank the reviewer for the suggestion. We could indeed list the on-rates in our table. However, an additional column would increase the width of the table to an extent that it would not fit on a single page anymore. We instead provided the formula for the straightforward calculation of the on-rate from the KD and the off-rate in the Supplementary Table 1 legend.

Technical issues: Every numerical result in the manuscript should include an experimental standard deviation. The authors state that the SPR, cell free degradation and intracellular degradation assays each were run in triplicate and MIC assays were run in duplicate. Yet for all of these assays they only report an average value. The authors must include an experimental standard deviation with every reported value from any of the assays so that a reader can judge the variability in a particular assay.

We added standard deviations to all reported values if applicable.

In the original manuscript, the authors stated that SPR assays were run in triplicate. However, in this updated data draft they now state that “At least two independent measurements were performed for each compound”. Why were some data sets left out in updated draft? The authors should add information to the Table which indicates the number of assays for a particular compound.

We thank the reviewer for pointing that out. During the revision process, we realized that for many compounds, only two measurements had been performed. We therefore corrected this statement and now indicate the number of measurements for each compound in Supplementary Table 1.

The authors should also clearly state what they used for the molar concentrations of the dimeric compounds compared to the monomeric forms.

We used the same molar concentrations for both monomeric compounds and Homo-BacPROTACs for the evaluation of all SPR experiments. We have now tried to describe our method more

precisely.

Since the authors are using molar concentration of compound, they should explain how doubling the concentration of the Cym binding sites for dimeric compounds will impact the on rate compared to the monomeric forms. All things being equal, would a 100 nM solution of monomer display the same apparent binding rate as a 100 nM solution of monomer or a 50 nM solution of dimer?

We have taken the decision to exclusively report molar concentrations of the full molecules regardless of the number of binding moieties. Indeed, 50 nM Homo-BacPROTAC would be expected to give the same signal as 100 nM monomer. For clarity, we have now included a respective remark in the legend of Supplementary Table 1. In addition, monomeric and dimeric compounds can be easily distinguished by color (green: Homo-BacPROTAC molecules), allowing the reader to take this into consideration when interpreting the data.

And any numerical result should be reported to the correct number of significant digits. For example, in Table 1 three compounds share identical MIC values at 0.097 uM. Is this MIC assay capable of nM resolution? How is it possible that three different compounds yield the exact same result? Two other compounds were reported with MIC values of 3.13 and two with 6.25. It seems odd that so many of the MIC values are the same.

We now report every numerical result with the correct number of significant digits.

Supplementary Tables 1-11

Supplementary Table 1. Structure-Activity Relationships part 1 (SPR, Caco-2)

compound ¹	Compound descriptor	K _b ClpC1-NTD ² [nM] Mean ± SD	K _b correlation ³	k _{off} ClpC1-NTD ⁴ [s ⁻¹] Mean ± SD	Caco-2 permeability ⁵				Permeability Correlation ⁶	
					P (A-B)	P (B-A)	Efflux ratio	Int. Perm	P (A-B)	Ratio
SI-38	[3-OPra]	5.6 ± 0.6	2	5.3E-02 ± 1.5E-02	<0.2	18		9.1	>75	>31
SI-64	[6-Me][3-OPra]	11.3 ± 1.1		1.6E-01 ± 1.5E-01	15	43	2.9	29.0		

It is always good to get in the habit of reporting experimental standard deviations and reporting them in a way that is meaningful to the reader. The number of significant digits is determined by the experimental standard deviation. For example, with the SPR data 5.6±0.6 is correct. But 11.3±1.1 should be reported as 11±1.

5.3E-02±1.5E-02 should be reported as 5E-02±2E-02

1.6E-01±1.5E-01 should be reported as 2E-01±2E-01 What does it mean to have an off rate with a standard error as big as the off rate itself? Does that tell us anything about results?

A minor point is the authors would save some clutter in the Table by not including the -0. None of their off rates need two units in the exp.

So, they could just write 5E-02±2E-02 as 5E-2±2E-2. Or better yet (5±2)E-2

We thank the reviewer for addressing this matter. We have implemented their suggestions in the Supplementary Table 1, which now looks less cluttered.

The authors should include an explanation as to why different amounts of dissociation phase data appear to be analyzed for different compounds. Many data sets seem to be fit out to 1000 seconds but others (typically the ones with a slower dissociation phase) seem to be missing dissociation phase data as highlighted below. Why were these data deleted?

8

9

10

11

12

13

During analysis of our SPR data, we noticed that the off-rates for some of the Homo-BacPROTACs showed some additional “sticky”/very slow off-component. This could be caused by dissociation of a Homo-BacPROTAC which crosslinks two ClpC1-NTD molecules on the chip. The example of compound SI-65 illustrates this (panel a below).

Of course, this behavior limits the applicability of the 1:1 binding model we used. We've actually observed this phenomenon in several PROTAC projects internally. It could be linked to various molecular properties like hydrophobicity of these relatively large molecules. In the case of Homo-PROTACs, avidity and rebinding events may further contribute to this phenomenon.

We generally evaluate the fit quality for a sensorgram based on two key characteristics: the U-value, which is implemented in the Biacore Insight Evaluation Software, "is a measure for the uniqueness of the calculated values for rate constants and Rmax. The U-value is determined by testing the dependence of the fit on correlated variations in pairs of parameters and is reported as a single value for the whole fitting. U-values above about 25 indicate that two or more of the parameters (rate constants and Rmax) are correlated and the absolute values cannot be determined. If the U-value is below about 15 the parameter values are not significantly correlated." (digi-33042-original (cytivalifesciences.com), access 19.10.2023, 13:22h) The second criterion for a "good" fit is the quality with which the fit represents the actual shape of the sensorgram based on visual inspection.

Forcing the software to fit non-1:1 data to a 1:1 model, in some cases, leads to fitting errors like mass-transport limited on-rates characterized by linear association phases (no curvature leading to loss of information) and premature saturation.

For cases, where the observed traces deviate from 1:1 behavior (most pronounced for the latter part of the dissociation phase), U-values are often on the high side. In this case, using only the earlier part of the dissociation phase often improves the fit as judged by the U-value.

We truncated dissociation phases for compounds displaying obvious deviation from 1:1 binding behavior in order to exclude the second slower dissociation phase and obtain the most accurate fit (judging by visual inspection and U-value, panel b).

a)

b)

Similarly, for monomers displaying fast dissociation, we decided to truncate the dissociation curve once the compound had fully dissociated to minimize artefacts caused by baseline drifts (see panels c and d, compound 20).

c)

d)

To make these specific quality control parameters of our sensorgrams more transparent to the reader, we have added a brief description of this part of the fitting procedure to the section 'SPR binding studies' in the Supplementary Information part.

REVIEWERS' COMMENTS

Reviewer #5 (Remarks to the Author):

No additional comments